# Intestinal interleukin-22 enhances GLP-1 production via the STAT3 pathway to improve glucose homeostasis during high-fat diet induced obesity in a study with male mice

Metabolic disorders such as obesity and diabetes are influenced by glucagon-like peptide-1 (GLP-1), which regulates insulin secretion. Interleukin (IL)−22 maintains intestinal barrier function, yet its role in metabolic regulation remains unclear. Here, we show that intestinal IL-22 deficiency reduces GLP-1 production and impairs glucose tolerance in high-fat diet−fed male mice, whereas long-term IL-22 administration restores GLP-1 levels, improves glucose tolerance, and normalizes insulin secretion and pancreatic islet size. IL-22 activates STAT3 binding to the *Gcg* promoter, indicating a direct role in GLP-1 induction. Butyrate supplementation increased IL-22 levels and enhanced GLP-1 production in an IL-22R−dependent manner, suggesting that microbial metabolites contribute to IL-22−mediated metabolic regulation. Direct IL-22 administration elevated circulating GLP-1 and improved glucose intolerance, while GLP-1 agonist treatment rescued metabolic defects associated with reduced IL-22 signaling. Conversely, the GLP-1 receptor antagonist exendin-9-39 abolished the glucose-lowering effects of IL-22, demonstrating that IL-22 acts primarily through GLP-1−dependent pathways. These findings identify IL-22 as an important regulator of intestinal GLP-1 production and glucose homeostasis during diet-induced obesity and highlight IL-22−GLP-1 signaling as a potential therapeutic axis for metabolic disorders.

The intestines play important roles in food digestion and absorption, and in immunity[1]. Some metabolic diseases are closely associated with intestinal disorders, which are affected by the intestinal hormones involved in metabolic regulation[2]. The oral ingestion of nutrients induces the secretion of incretin hormones[3,4], including glucagon-like peptide-1 (GLP-1) and other glucose-dependent insulinotropic polypeptides. GLP-1 is encoded by the preproglucagon gene. This gene is expressed in L cells of the distal jejunum, ileum, and colon. Additionally, it is expressed in pancreatic alpha cells and some neurons[5,6]. GLP-1 is a glucose-dependent insulinotropic hormone that also promotes β-cell proliferation[7]. GLP-1 receptor agonizts effectively lower blood glucose levels effectively and improve glycemic control in patients with type 2 diabetes[8]. However, immune cell-mediated signaling factors that trigger GLP-1 release, besides the inflammatory cytokine IL-6, remain largely understudied[9].

✉e-mail: cchung@yonsei.ac.kr; hjko@kangwon.ac.kr

Obesity is a major risk factor for metabolic syndromes such as diabetes mellitus and metabolic dysfunction–associated liver disease[10,11]. The Western diet is characterized by high fat content. This increases insulin resistance and glucose intolerance, consequently leading to obesity. In addition, a high-fat diet (HFD) affects the composition of the gut microbiota. This imbalance in microbial communities exacerbates insulin resistance and glucose intolerance, leading to metabolic disorders such as diabetes mellitus[12]. Metabolites of the gut microbiota, such as short-chain fatty acids (SCFAs), contribute to the regulation of host energy metabolism and prevention of diseases such as obesity and diabetes mellitus[13]. Additionally, the transplantation of fecal microbiota ameliorates insulin resistance in humans[14]. However, the role of intestinal immune cells in the metabolic regulation associated with obesity remains unclear.

IL-22, a member of the IL-10 cytokine superfamily, regulates immune response and maintains tissue homeostasis[15]. It protects the intestinal mucosa, promotes the survival and regeneration of intestinal cells, and contributes to the regulation of inflammatory responses[16]. This cytokine has beneficial effects against inflammatory bowel diseases and other metabolic syndromes[17–19]. IL-22 levels are decreased in patients with obesity[20]. Similarly, insulin resistance increased in mice lacking IL-22R1 and improved after IL-22 administration[20]. However, it remains unclear how IL-22 prevents and alleviates obesity-related metabolic disorders, beyond its established role in mucosal immune regulation.

Here, we show that IL-22 influences intestinal GLP-1 production in obese mice subjected to an HFD. We further show that IL-22 is associated with improved glucose tolerance under these conditions. Our findings indicate that IL-22 mediates intestinal GLP-1 production, thereby contributing to the maintenance of glucose homeostasis in obesity. These results suggest that modulation of IL-22 signaling may have relevance for obesity-related metabolic disorders.

## Results

### Correlation between GLP-1 and IL-22 levels in intestinal homeostasis

We evaluated the intestinal levels of GLP-1 and IL-22 in mice subjected to a regular diet (RD) or HFD for 12 weeks. Mice on the HFD exhibited a significant reduction in intestinal GLP-1 levels compared to those on an RD (Fig. 1a). A substantial reduction in intestinal IL-22 levels was observed in mice consuming HFD following the decrease in GLP-1 level (Fig. 1b). Moreover, a positive correlation was observed between intestinal GLP-1 and IL-22 levels in both groups (Fig. 1c). Next, we assessed the proportion of RORγt[+] ILCs in the lamina propria, a major IL-22-producing cell population. However, no significant difference was observed between RD- and HFD-fed mice (Fig. 1d). The proportion of RORγt[+] CD4[+] T cells decreased in HFD-fed mice compared to those on an RD (Fig. 1e). In addition, the proportion of IL-22[+] ILCs and IL-22[+] CD4 T cells was reduced in HFD-fed mice (Fig. 1f, g). The gating strategy is shown in Supplementary Fig. 1.

CCR6 is highly expressed in IL-22-producing cells, such as ILC3s, and CCR6 deficiency reduces IL-22 production[21]. Therefore, we used CCR6[−/−] mice to investigate the effect of decreased IL-22 production caused by CCR6 deficiency on intestinal GLP-1 levels. CCR6-deficient

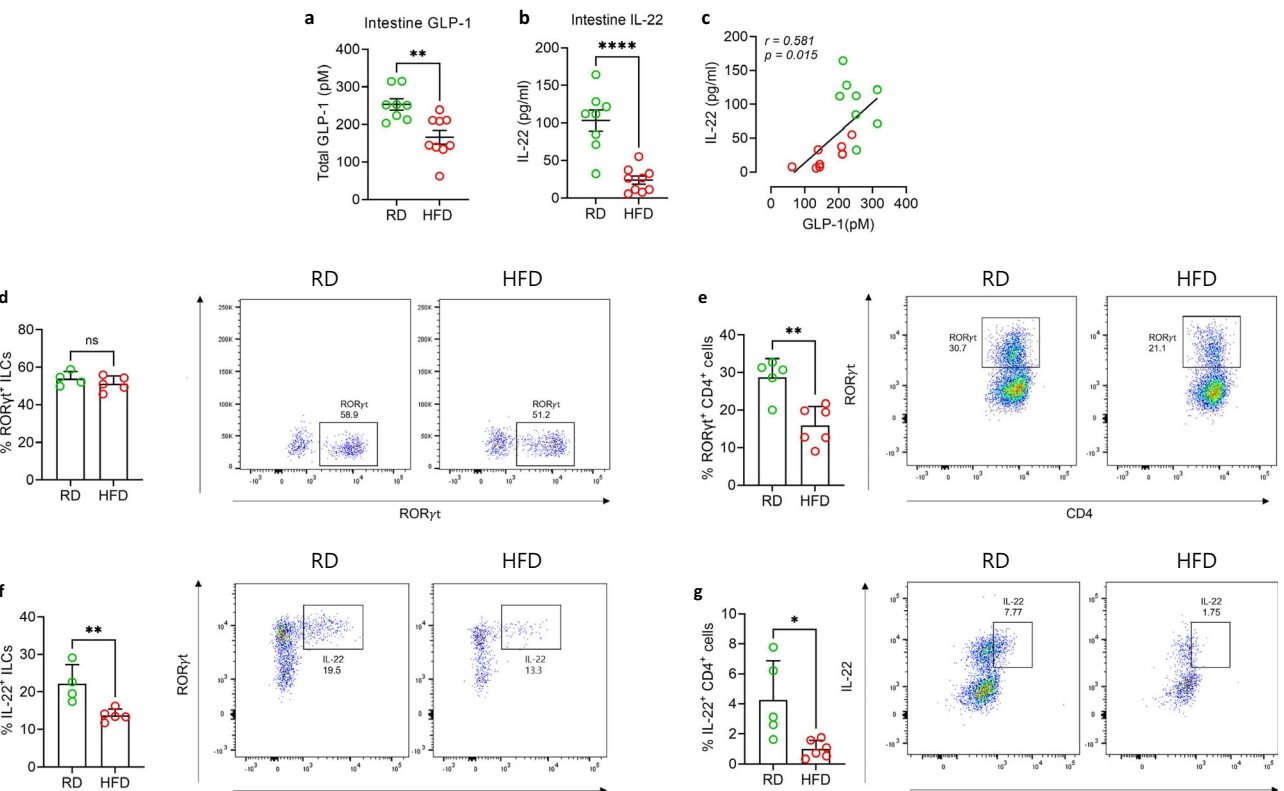

**Fig. 1 | Obesity decreased GLP-1 and IL-22 production in the gut.** C57BL/6 mice were fed a regular (RD) or high-fat (HFD) diet for 12 weeks. The level of (**a**) GLP-1 (P = 0.002) and (**b**) IL-22 (P < 0.0001) in ileum tissue homogenates of RD (n = 8) or HFD (n = 9)−fed mice is shown. **c** A scattered plot of GLP-1 and IL-22 levels in the ileum tissue of RD- (red) and HFD-fed (green) mice. Flow cytometric analysis of laminar propria (LP) cells isolated from mice fed a regular (RD) and high-fat (HFD) diet. **d** The proportion of RORγt[+] ILCs. **e** The proportion of RORγt[+] CD4[+] T cells (P = 0.002). **f** The proportion of IL-22[+] ILCs (P = 0.009). **g** The proportion of IL-22[+] CD4[+] T cells (P = 0.01). For flow cytometric analyses, n = 4 (RD) and n = 5 (HFD) for (**d**, **f**), and n = 5 (RD) and n = 6 (HFD) for (**e**, **g**). Each data point represents one biologically independent mouse. Data in (**a**–**c**) are representative of three independent experiments, and data in (**d**–**g**) are representative of two independent experiments. Statistical significance was analyzed using an unpaired two-tailed Student's t-test (**a**, **b**, **d**–**g**). Correlation was assessed using Pearson's correlation coefficient with a two-tailed test (**c**). P < 0.05 (*), P < 0.01 (**), P < 0.0001 (****). ns not significant. Data are presented as mean ± SEM.

mice exhibited lower intestinal IL-22 levels than WT mice (Supplementary Fig. 2a). In addition, HFD-fed CCR6$^{-/-}$ mice exhibited worsened glucose intolerance compared to WT mice (Supplementary Fig. 2b, c); no insulin resistance was observed (Supplementary Fig. 2d, e). GLP-1 levels and GLP-1-expressing cells in the intestine (Supplementary Fig. 2f–h) were also reduced following the decrease in intestinal IL-22 levels. Additionally, serum insulin concentrations and pancreatic islet size were reduced in CCR6$^{-/-}$ mice (Supplementary Fig 2i–k). We next evaluated the effects of exogenous IL-22 administration in CCR6$^{-/-}$ mice. Long-term IL-22 administration did not significantly affect body weight in CCR6$^{-/-}$ mice (Supplementary Fig. 3a). However, IL-22 treatment markedly improved glucose tolerance, as evidenced by reduced blood glucose levels during the glucose tolerance test (Supplementary Fig. 3b). IL-22 also increased the number of GLP-1$^+$ L cells in the intestine of HFD-fed CCR6$^{-/-}$ mice (Supplementary Fig. 3c, d). Similarly, chronic IL-22 administration increased GLP-1$^+$ L cell abundance in HFD-fed WT mice (Supplementary Fig. 3e, f). IL-22 also elevated circulating insulin levels compared with vehicle-treated CCR6$^{-/-}$ mice (Supplementary Fig. 3g). Moreover, histological and immunofluorescence analyses revealed that IL-22 restored pancreatic islet size in HFD-fed CCR6$^{-/-}$ mice (Supplementary Fig. 3h). These findings demonstrate the role of IL-22 in GLP-1 regulation and maintenance of metabolic health.

We conducted transcriptomic analysis to compare gene expression in human metabolic diseases based on data sourced from the GSE database. The expression levels of *GCG* in ileal biopsy tissues were significantly decreased in patients with diabetes compared to non-diabetic patients with obesity in the GSE132831 dataset (Supplementary Fig. 4a). Additionally, the expression of IL-22-related genes markedly decreased in patients with comorbid diabetes and obesity (Supplementary Fig. 4b). Correlation analysis of the expression levels of these genes across patients revealed a positive correlation between *GCG* and IL-22-related genes (Supplementary Fig. 4c). The association between these genes is shown revealing downregulated expression in patients with obesity who have diabetes. (Supplementary Fig. 4d). Furthermore, analysis of the GSE193677 dataset confirmed a positive correlation between *GCG* and IL-22-related genes in the gut of healthy individuals (Supplementary Fig. 4e). These results suggest a potential association between GLP-1 and IL-22 in humans and in murine models.

## Impaired intestinal IL-22 signaling exacerbates metabolic syndrome by reducing GLP-1 production

IL-22 signaling in the intestine maintains intestinal homeostasis and immunity[22]. In this study, the IL-22 receptor (IL-22R) was highly expressed in human intestines, as determined using the GSE125970 dataset (Supplementary Fig. 5). We further investigated whether IL-22 signaling, inhibited by IL-22R blockade, affects obesity-related metabolic changes via GLP-1 production. We generated IL-22RA1$^{Vil KO}$ mice by crossing Villin cre mice with IL-22RA1$^{(f/f)}$ mice to achieve the ablation of IL-22 signaling in intestinal epithelial cells. IL-22RA1$^{(f/f)}$ and IL-22RA1$^{Vil KO}$ mice were fed an HFD for 12 weeks. There was no significant difference in body weight between HFD-fed IL-22RA1$^{Vil KO}$ and IL-22RA1$^{(f/f)}$ mice (Fig. 2a). Additionally, no difference in food intake was observed between the two groups (Supplementary Fig. 6a). Oral glucose tolerance test (OGTT) revealed exacerbated glucose intolerance in IL-22RA1$^{Vil KO}$ mice compared to IL-22RA1$^{(f/f)}$ mice (Fig. 2b, c). A similar impairment was also observed during the intraperitoneal glucose tolerance test (Supplementary Fig. 6b, c). Following oral glucose administration, IL-22RA1$^{Vil KO}$ mice exhibited markedly reduced active GLP-1 level compared with IL-22RA1$^{(f/f)}$ mice (Fig. 2d). Consistent with impaired GLP-1 production, IL-22RA1$^{Vil KO}$ mice exhibited significantly reduced insulin levels at the 15-min time point in both oral and intraperitoneal glucose administration. (Fig. 2e, f). IL-22RA1$^{Vil KO}$ mice exhibited insulin resistance, as indicated by impaired control of blood glucose levels (Fig. 2g, h).

We hypothesized that the absence of IL-22 signaling affects GLP-1 production. Consistent with this hypothesis, intestinal GLP-1 levels were significantly reduced in IL-22RA1$^{Vil KO}$ mice compared to IL-22RA1$^{(f/f)}$ mice (Fig. 2i). Additionally, the transcriptional levels of *RORγt* (Fig. 2j) and *Gcg* (Fig. 2k) were significantly lower in the small intestine of HFD-fed IL-22RA1$^{Vil KO}$ mice than in HFD-fed IL-22RA1$^{(f/f)}$ mice. The expression of the tight junction proteins Occludin (Supplementary Fig. 6d) was reduced in HFD-fed IL-22RA1$^{Vil KO}$ mice.

GLP-1 contributes to blood glucose regulation by inducing insulin release. Therefore, we assessed serum insulin concentrations in IL-22RA1$^{(f/f)}$ and IL-22RA1$^{Vil KO}$ mice. In addition to decreased GLP-1 production, serum insulin concentrations were also significantly low in HFD-fed IL-22RA1$^{Vil KO}$ mice (Fig. 2l). Furthermore, immunofluorescent staining revealed a reduction in insulin staining in HFD-fed IL-22RA1$^{Vil KO}$ mice (Fig. 2m, n). Similarly, H&E staining showed a decrease in the area of islets in the pancreas (Fig. 2o, p). Fasting glucagon levels were similar between IL-22RA1$^{Vil KO}$ and control mice, indicating that their impaired glucose tolerance is not due to reduced glucagon (Fig. 2q). We next assessed systemic energy metabolism in HFD-fed IL-22RA1$^{(f/f)}$ and IL-22RA1$^{Vil KO}$ mice. Energy expenditure was comparable between IL-22RA1$^{(f/f)}$ and IL-22RA1$^{Vil KO}$ during both light and dark cycles (Supplementary Fig. 7a, b). Respiratory exchange ratio (RER) was also similar between the two groups (Supplementary Fig. 7c, d). Likewise, oxygen consumption (VO$_2$) and carbon dioxide production (VCO$_2$) did not differ between IL-22RA1$^{(f/f)}$ and IL-22RA1$^{Vil KO}$ mice (Supplementary Fig. 7e–h).

We further aimed to determine whether IL-22 signaling deficiency under homeostatic conditions in intestine epithelial cells induces metabolic disorders. No differences in body weight changes (Supplementary Fig. 8a) or food intake (Supplementary Fig. 8b) were observed between IL-22RA1$^{(f/f)}$ and IL-22RA1$^{Vil KO}$ mice fed an RD for 12 weeks. The intraperitoneal glucose tolerance test revealed no exacerbation of glucose intolerance attributable to the deficiency of IL-22 signaling (Supplementary Fig. 8c, d). Similarly, no differences in insulin sensitivity were observed between the two groups (Supplementary Fig. 8e, f). We subsequently evaluated GLP-1 levels in the intestines. Mice fed an RD exhibited comparable levels of intestinal GLP-1 (Supplementary Fig. 8g) and *Gcg* expression (Supplementary Fig. 8h).

We generated mice lacking IL-22 receptors in Gcg-expressing cells by crossing Gcg cre mice with IL-22RA1$^{(f/f)}$ mice to evaluate the levels of intestinal GLP-1 and insulin in the absence of IL-22 signaling in L cells. IL-22RA1$^{(f/f)}$ and IL-22RA1$^{Gcg KO}$ mice were fed an HFD for 12 weeks. Although the body weights of these mice did not significantly differ (Fig. 3a), HFD-fed IL-22RA1$^{Gcg KO}$ mice exhibited worse glucose intolerance than IL-22RA1$^{(f/f)}$ mice in OGTT (Fig. 3b, c). This impairment was similarly observed in the intraperitoneal glucose tolerance test (Supplementary Fig. 9a, b). Following oral glucose administration, active GLP-1 levels were lower in IL-22RA1$^{Gcg KO}$ mice compared with IL-22RA1$^{(f/f)}$ mice (Fig. 3d). Consistent with reduced GLP-1 level, insulin levels after oral glucose administration were markedly decreased in IL-22RA1$^{Gcg KO}$ mice (Fig. 3e) However, this difference was not statistically significant during the intraperitoneal glucose administration (Fig. 3f). Furthermore, no difference in insulin resistance was observed between the two groups (Fig. 3g, h). Next, we investigated whether the absence of IL-22 signaling in L cells reduces intestinal GLP-1 production. Intestinal GLP-1 levels (Fig. 3i) and the number of L cells in the small intestine was reduced in IL-22RA1$^{Gcg KO}$ mice, as determined by counting GLP-1−expressing cells (Fig. 3j, k). Furthermore, transcriptional levels of intestinal *RORγt* (Fig. 3l) and *Gcg* (Fig. 3m) decreased in HFD-fed IL-22RA1$^{Gcg KO}$ mice. In addition, serum insulin levels decreased in HFD-fed IL-22RA1$^{Gcg KO}$ mice (Fig. 3n). The insulin-positive area and pancreatic islets size were also significantly reduced in IL-22RA1$^{Gcg KO}$ mice, as shown by immunofluorescence and histological analyses (Fig. 3o–r). Fasting glucagon levels did not differ between IL-22RA1$^{Gcg KO}$ and control

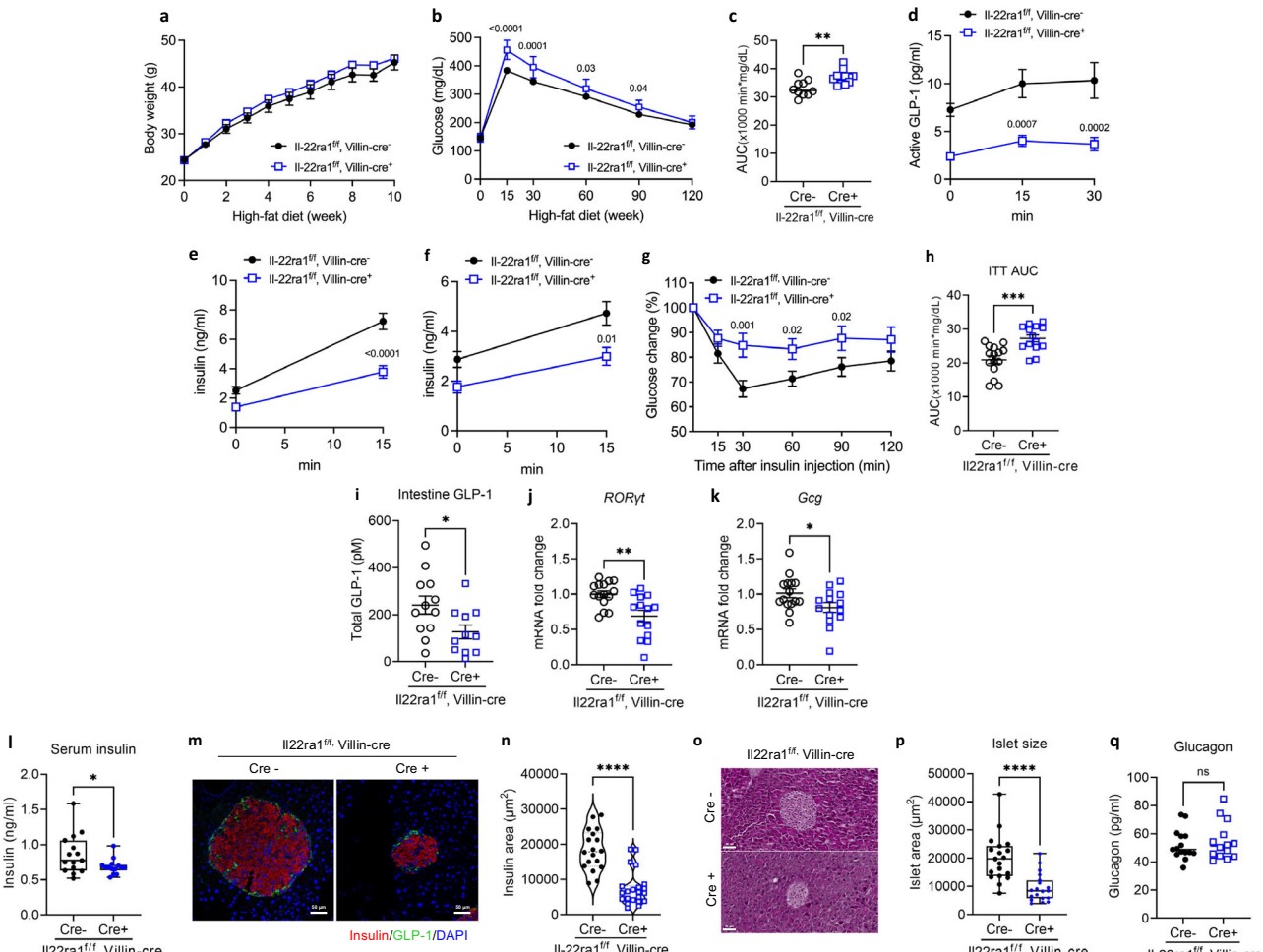

**Fig. 2 | Impaired IL-22 signaling in intestinal epithelial cells induces glucose intolerance.** IL-22RA1[(f/f)] (*n* = 15) and IL-22RA1[vil KO] (*n* = 14) mice were fed a HFD for 12 weeks. **a** Weekly bodyweight changes of IL-22RA1[(f/f)] (*n* = 10) and IL-22RA1[vil KO] mice (*n* = 12). **b** Oral glucose (1 g/kg) tolerance test (OGTT) in 16 h-fasted mice at 8 weeks. **c** The area under the curve (AUC) during OGTT (*P* = 0.001). **d** Plasma Active GLP-1 levels following oral glucose administration (0, 15, 30 min) in IL-22RA1[(f/f)] (*n* = 14) and IL-22RA1[vil KO] mice (*n* = 11). **e** Plasma insulin levels following oral glucose administration (0, 15 min) in IL-22RA1[(f/f)] (*n* = 14) and IL-22RA1[vil KO] mice (*n* = 11). **f** Plasma insulin levels following intraperitoneal glucose administration (0, 15 min) in IL-22RA1[(f/f)] (*n* = 14) and IL-22RA1[vil KO] mice (*n* = 11). **g** Insulin (1 U/kg) tolerance tests (ITT) in 6-h-fasted mice at 10 weeks. **h** The AUC during ITT (*P* = 0.0003). **i** The levels of GLP-1 from the mouse intestinal homogenate (*P* = 0.02). mRNA levels of (**j**) *RORγt* (*P* = 0.002) and (**k**) *Gcg* (*P* = 0.03) in the small

intestines. **l** Serum insulin levels in IL-22RA1[(f/f)] and IL-22RA1[vil KO] mice (*P* = 0.03). **m** Immunofluorescence staining image for insulin (red), GLP-1 (green), and nuclei (DAPI, blue) extracted from the mouse pancreas. Original magnification: 20× (scale bar, 50 μm). **n** Quantification of insulin-positive area (*P* < 0.0001). **o** H&E staining of the mouse pancreas. Original magnification: 20× (scale bar, 50 μm). **p** Quantification of pancreatic islet size (*P* < 0.0001). **q** Fasting glucagon levels. Statistical significance was analyzed using two-way ANOVA (**b**, **d**, **g**) with fisher's LSD test for multiple comparisons, and unpaired two-tailed Student's t-test (**c**, **e**, **f**, **h**, **l**, **n**, **p**, **q**). *P* < 0.05 (*), *P* < 0.01 (**), *P* < 0.001 (***), *P* < 0.0001 (****). ns not significant. Data are presented as mean ± SEM. Box plots show the median (center line), the 25th and 75th percentiles (box), and the minimum and maximum values (whiskers). Representative data are shown from three independent experiments.

mice (Fig. 3s). We next evaluated metabolic parameters in HFD-fed IL-22RA1[(f/f)] and IL-22RA1[Gcg KO] mice. No differences in energy expenditure were observed between the two groups across both light and dark periods (Supplementary Fig. 10a, b). The respiratory exchange (RER) also remained comparable between genotypes (Supplementary Fig. 10c, d). Similarly, oxygen consumption (VO₂) and carbon dioxide production (VCO₂) were not different between IL-22RA1[(f/f)] and IL-22RA1[Gcg KO] mice (Supplementary Fig. 10e–h).

We fed IL-22RA1[(f/f)] and IL-22RA1[Gcg KO] mice an RD for 12 weeks to identify metabolic defects in the absence of IL-22 in L cells under normal dietary conditions. No significant differences were observed in body weight changes (Supplementary Fig. 11a) or food intake (Supplementary Fig. 11b) between the two groups. Moreover, no exacerbation of glucose intolerance was attributable to IL-22 signaling deficiency (Supplementary Fig. 11c, d). Similarly, no differences in insulin sensitivity were observed between the groups

(Supplementary Fig. 11e, f). We then assessed the levels of GLP-1 in the intestines. Mice on an RD showed similar levels of intestinal GLP-1 (Supplementary Fig. 11g) and *Gcg* expression (Supplementary Fig. 11h). Serum insulin levels were also similar between the groups (Supplementary Fig. 11i). Additionally, immunofluorescence staining in IL-22RA1[(f/f)] and IL-22RA1[Gcg KO] mice fed an RD showed comparable insulin staining (Supplementary Fig. 11j), whereas H&E staining revealed comparable pancreatic islet areas between these groups (Supplementary Fig. 11k, l).

## IL-22 induced GLP-1 production in L cells
In this study, we investigated whether IL-22 directly controls GLP-1 production in L cells. IL-22 increased the production of GLP-1 (Fig. 4a) in a dose-dependent manner (Fig. 4b) in STC-1 cells. As GLP-1 production in L cells relies on calcium signaling, we evaluated the effects of nifedipine, a calcium channel blocker. The inhibition of calcium

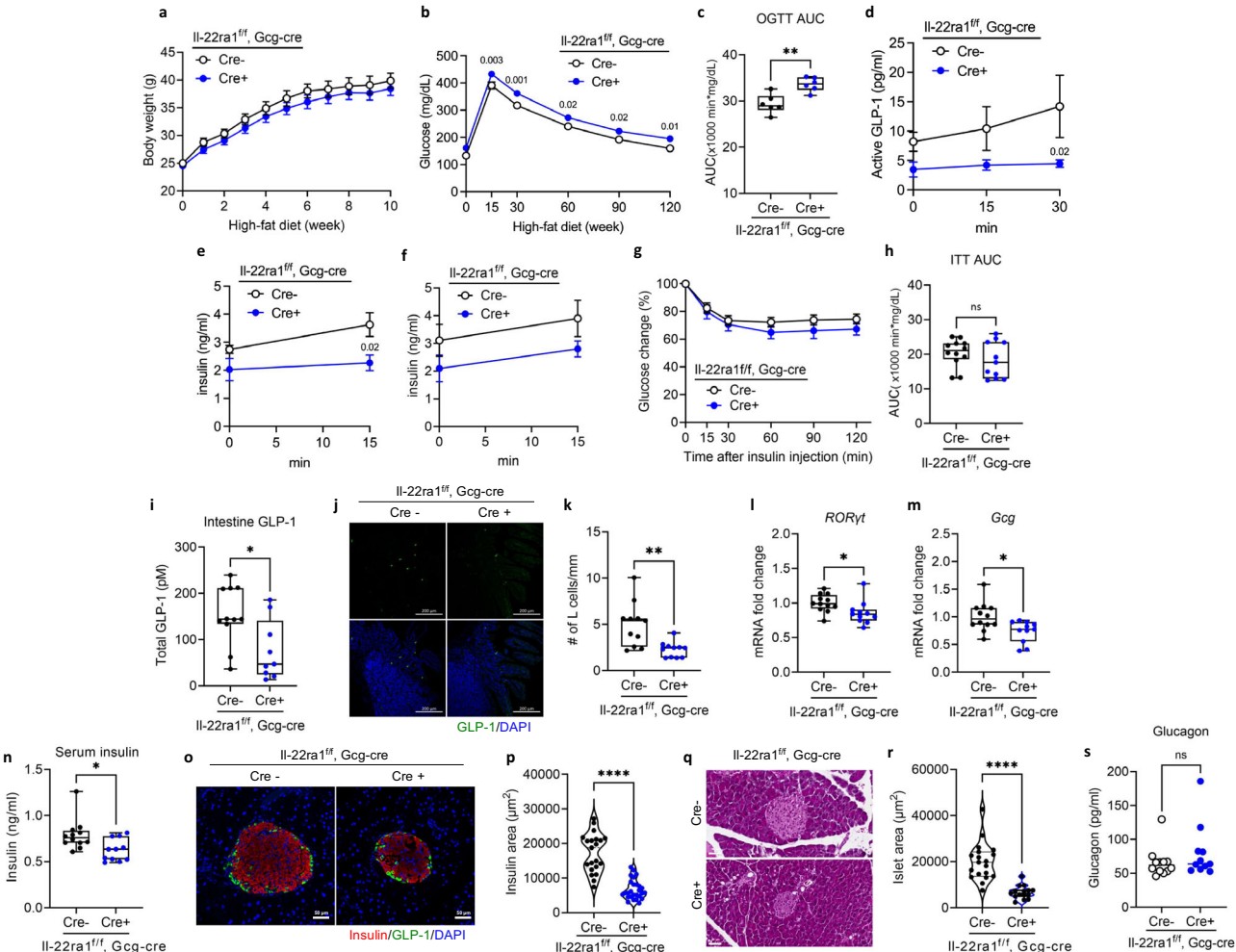

**Fig. 3 | Ablation of Gcg-specific IL-22 signaling exacerbates glucose intolerance.**
IL-22RA1[(f/f)] ($n = 12$) and IL-22RA1[Gcg KO] ($n = 11$) mice were fed a HFD for 12 weeks.
**a** Weekly bodyweight changes in IL-22RA1[(f/f)] and IL-22RA1[Gcg KO] mice. **b** OGTT (1 g/
kg) in 16-h-fasted mice at 8 weeks. **c** The AUC during OGTT ($P = 0.002$). **d** Plasma
Active GLP-1 levels following oral glucose administration (0, 15, 30 min) in IL-
22RA1[(f/f)] ($n = 6$) and IL-22RA1[Gcg KO] mice ($n = 6$). **e** Plasma insulin levels following oral
glucose administration (0, 15 min) in IL-22RA1[(f/f)] ($n = 6$) and IL-22RA1[Gcg KO] mice
($n = 6$). **f** Plasma insulin levels following intraperitoneal glucose administration (0,
15 min) in IL-22RA1[(f/f)] ($n = 6$) and IL-22RA1[Gcg KO] mice ($n = 6$). **g** ITT in 6-h-fasted mice
at 10 weeks. **h** The AUC during ITT. **i** The levels of GLP-1 extracted from the mouse
intestinal supernatant ($P = 0.01$). **j** Immunofluorescence staining image for GLP-1
(green) and nuclei (DAPI, blue) from the mouse intestines. Original magnification:
20× (scale bar, 200 μm). **k** The number of L cells in the intestines ($P = 0.003$). mRNA

levels of (**l**) *RORγt* ($P = 0.03$) and (**m**) *Gcg* ($P = 0.01$) in the small intestines. **n** Serum
insulin levels in IL-22RA1[(f/f)] and IL-22RA1[Gcg KO] mice ($P = 0.01$).
**o** Immunofluorescence staining image for insulin (red), GLP-1 (green), and nuclei
(DAPI, blue) extracted from the mouse pancreas. Original magnification: 20× (scale
bar, 50 μm). **p** Quantification of insulin-positive area ($P < 0.0001$). **q** H&E staining of
the mouse pancreas. Original magnification: 20× (scale bar, 50 μm). **r** Quantification
of pancreatic islet size ($P < 0.0001$). **s** Fasting glucagon levels. Statistical sig-
nificance was analyzed using two-way ANOVA (**b**, **d**) with fisher's LSD test for
multiple comparisons, and unpaired two-tailed Student's t-test (**c**, **e**, **f**, **h**, **I**, **k**–**n**,
**p**, **r**, **s**). $P < 0.05$ (*), $P < 0.01$ (**), $P < 0.001$ (***), $P < 0.0001$ (****). ns not significant.
Data are presented as mean ± SEM. Box plots show the median (center line), the
25th and 75th percentiles (box), and the minimum and maximum values (whiskers).
Representative data are shown from three independent experiments.

signaling reduced GLP-1 levels via IL-22 in STC-1 cells (Fig. 4c). These
results suggest that IL-22 contributes to GLP-1 induction via calcium
signaling in L cells. Furthermore, the transcript levels of *Gcg* and *Ffar2*
increased following IL-22 treatment in STC-1 cells (Supplementary
Fig 12a).

IL-22 exerts its functions via STAT3 activation. Therefore, we
assessed the effect of a STAT3 inhibitor on IL-22-induced GLP-1 pro-
duction in STC-1 cells. The STAT3 inhibitor attenuated the IL-22-
induced increase in GLP-1 level, suggesting that STAT3 signaling is
involved in IL-22-mediated GLP-1 production (Fig. 4d). The STAT3
binding sequence TT (N4-6) AA was identified in the glucagon pro-
moter region to confirm the binding of IL-22-induced STAT3 to this
promoter (Supplementary Fig. 12b). A putative STAT3 binding site was
predicted between nucleotides -1058 and -1068 of the promoter
region. We identified STAT3 binding sites within the mouse glucagon

promoter and validated the direct binding of STAT3 using ChIP.
Moreover, the treatment of STC-1 cells with IL-22 further increased
STAT3 binding to the glucagon promoter (Fig. 4e).

We treated small intestinal organoids derived from Gcg-Cre;
Rosa-YFP mice with IL-22. The number of YFP-expressing L cells
increased in response to IL-22 treatment. However, this increase was
attenuated upon co-treatment with a STAT3 inhibitor (Fig. 4f, g).
Immunofluorescence staining revealed an increase in YFP expression
in organoids derived from Gcg-Cre; Rosa-YFP mice treated with
10 ng/ml IL-22 for two days compared to those treated with the
vehicle (Fig. 4h). Additionally, we confirmed a dose-dependent
increase in supernatant GLP-1 levels (Fig. 4i) and *Gcg* mRNA expres-
sion levels in response to IL-22 (Fig. 4j). These findings indicate that
IL-22 activates STAT3. The activated STAT3 binds to the glucagon
promoter to increase the expression of glucagon, which is further

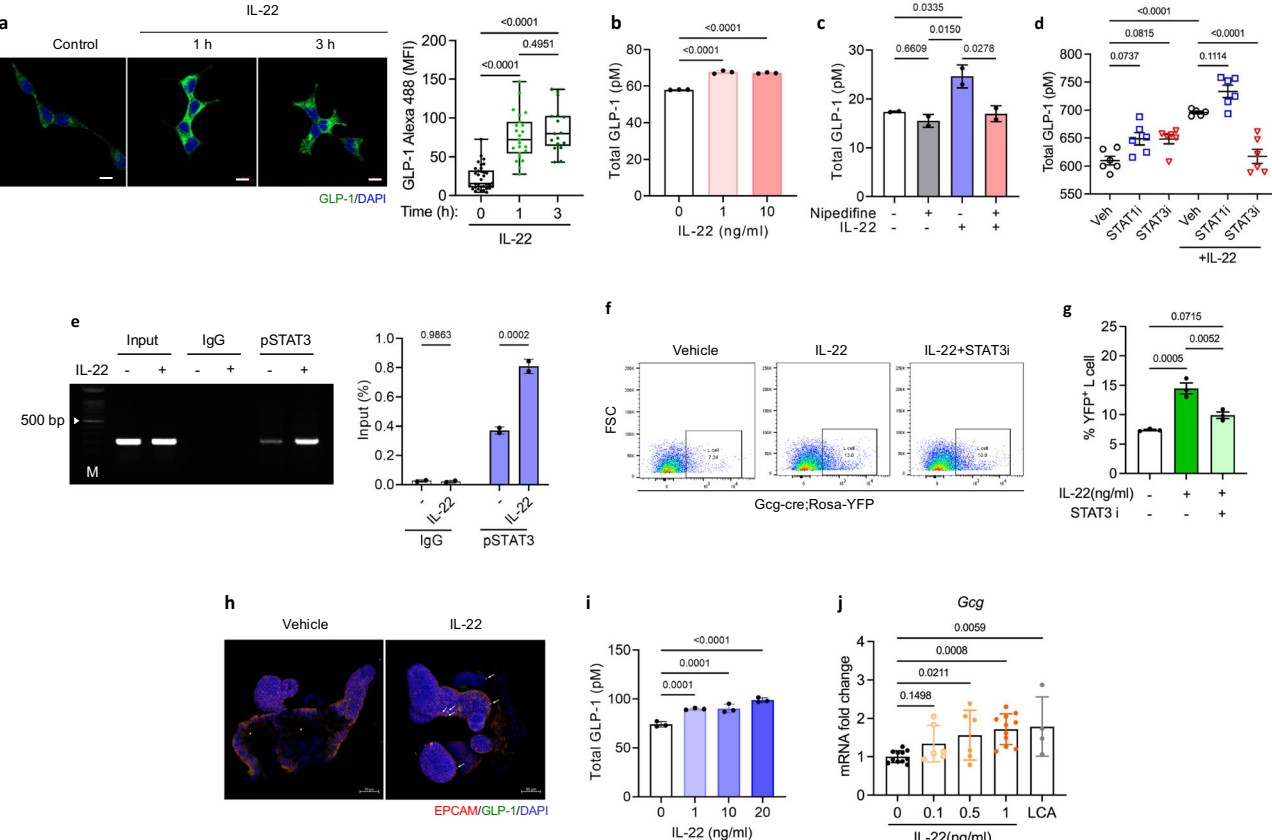

**Fig. 4 | IL-22 induces GLP-1 through the STAT3 pathway. a** Immunofluorescence staining of GLP-1 in STC-1 cells treated with recombinant mouse IL-22 (10 ng/ml). Original magnification: 400× (scale bar, 10 μm). Quantification of GLP-1 intensity (Alexa 488) is presented as a box plot. **b** Supernatant GLP-1 levels in STC-1 cells treated with varying concentrations of IL-22 (0, 1, 10, and 100 ng/ml). **c** Supernatant GLP-1 levels in STC-1 cells treated with IL-22 (10 ng/ml) with or without nifedipine (10 μM). **d** Supernatant GLP-1 levels in STC-1 cells treated with a STAT1 or STAT3 inhibitor (10 μm), followed by IL-22 (10 ng/mL) treatment. **e** Immunoprecipitated DNA amplified via PCR using primers corresponding to the Gcg promoter region containing the STAT3 binding site for ChiP analysis. Quantitative ChiP analysis. **f** FACS analysis of Gcg-cre-ROSA-YFP mouse intestinal organoids treated with IL-22 (1 ng/ml) and a STAT3 inhibitor (100 ng/ml) for 72 h. **g** Quantification of YFP-positive cells. **h** Immunofluorescence staining of Gcg-cre-ROSA-YFP mouse intestinal organoids treated with IL-22 (10 ng/ml), showing EPCAM (red), GLP-1 (green), and nuclei (DAPI, blue). Original magnification: 20× (scale bar, 50 μm). Data was obtained using organoids derived from four independent mice. **i** Supernatant GLP-1 levels in intestinal organoids treated with varying concentrations of IL-22 (0, 1, 10, and 20 ng/ml). **j** mRNA expression levels of *Gcg* in Gcg-cre-ROSA-YFP mouse intestinal organoids treated with IL-22 (0, 0.1, 0.5, and 1 ng/ml) or lithocholic acid (10 μM). Statistical significance was analyzed using Ordinary one-way ANOVA (**a–e**, **g**, **i**, **j**) with fisher's LSD test for multiple comparisons. $P < 0.05$ (*), $P < 0.01$ (**), $P < 0.001$ (***), $P < 0.0001$ (****). ns not significant. Data are presented as mean ± SEM. Box plots show the median (center line), the 25th and 75th percentiles (box), and the minimum and maximum values (whiskers). Representative data are shown from two (**a–e**)–three (**f–j**) independent experiments.

converted into GLP-1 under catalysis by the activated prohormone convertase 1/3.

## Intestinal IL-22 production is impaired by an HFD through alteration of microbial composition

We evaluated the potential effects of changes in microbial diversity under HFD-induced obesity on immune function and metabolic regulation. We compared the microbial communities in fecal samples from mice fed an RD and those on an HFD to investigate whether the reduction of IL-22 in obese mice is related to changes in the gut microbiota. Our results revealed distinct compositional changes in the overall gut microbiota between the two groups. At the genus level, a noticeable decrease in the abundance of *Lachnospiraceae_NK4A136* and *Muribaculaceae* was observed (Supplementary Fig. 13a). Obese mice exhibited a significant decrease in alpha diversity indices, including the Shannon, Gini-Simpson, and ACE indices, compared to RD mice (Supplementary Fig. 13b–d). This suggests that an HFD may contribute to a reduction in the diversity and richness of the gut microbiota. Distinct clustering in the microbial composition between mice fed an RD and those fed an HFD is shown in (Supplementary Fig. 13e–g). The clear separation in beta diversity between the RD and

HFD groups indicates that the significant differences in the microbial community structure can be attributed to dietary composition. The abundance of bacteria producing SCFAs, including butyrate, was reduced in HFD-fed mice compared to RD-fed mice (Supplementary Fig. 13h–l). Measurements of SCFA levels in fecal samples from both groups indicated a decrease in the concentration of SCFAs in the feces of HFD-fed mice (Supplementary Fig. 13m–o). A butyrate concentration-dependent increase in IL-22 expression was observed in the supernatant of mouse lamina propria cells treated with butyric acid (Supplementary Fig. 13p). Furthermore, a positive correlation was observed between the abundance of SCFA-producing bacteria and intestinal IL-22 or GLP-1 levels (Supplementary Fig. 13q, r) during obesity. These results suggest that the dysregulation of the gut microbiota, particularly the reduction in the abundance of butyrate-producing bacteria, contributes to the reduction in IL-22 levels under obesity. This may lead to a decrease in GLP-1 levels.

Butyrate induces IL-22 expression and alleviates insulin resistance in obese mouse models[23,24]. Therefore, we investigated whether metabolic defects in IL-22RA1[(f/f)] and IL-22RA1[Vil KO] mice could be ameliorated by IL-22-inducing butyrate supplementation. Butyrate supplementation showed no significant differences in body weight

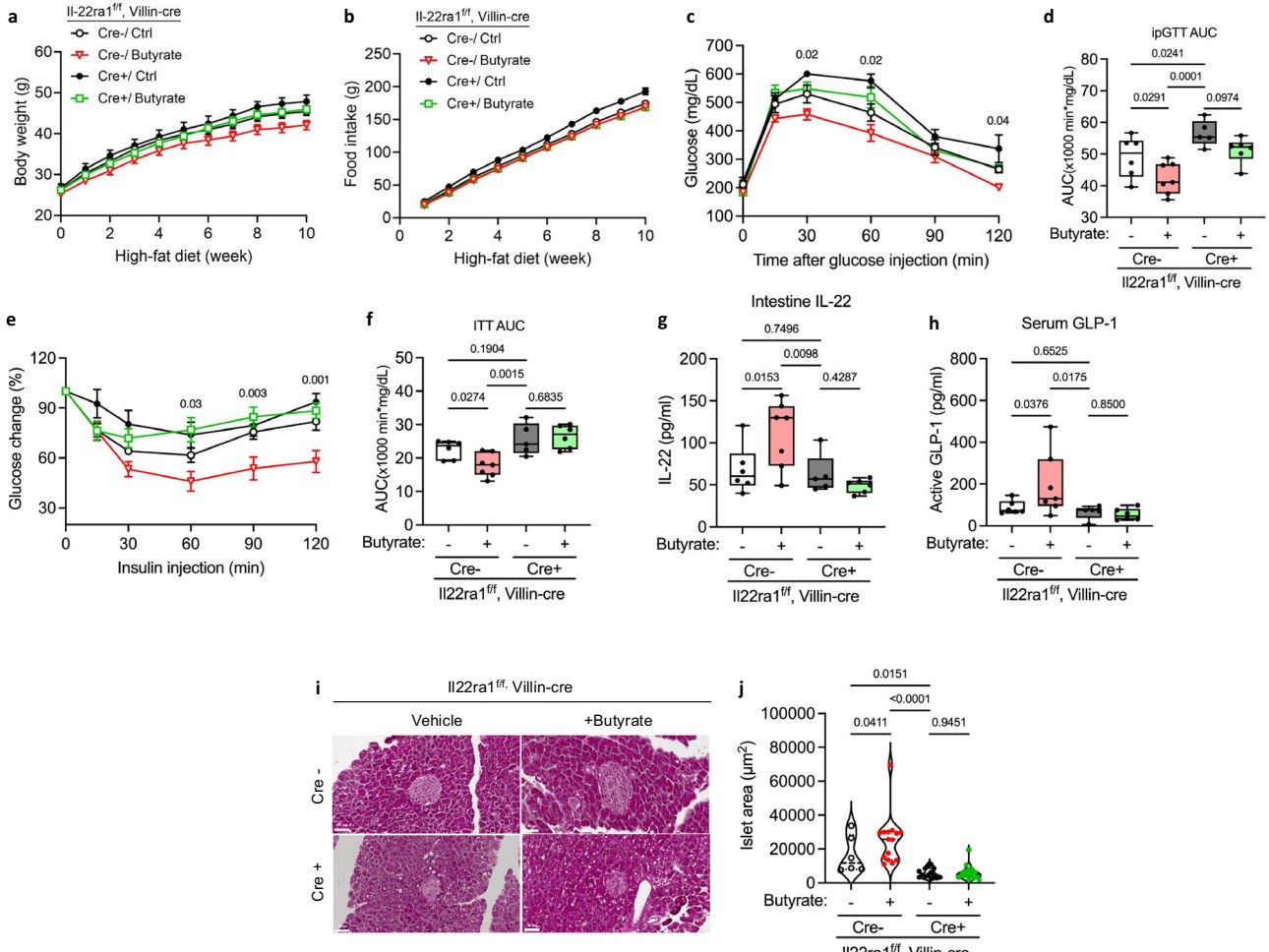

**Fig. 5 | Butyrate activated by IL-22 alleviates metabolic syndrome.** IL-22RA1$^{(f/f)}$ and IL-22RA1$^{Vil\ KO}$ mice were fed an HFD for 12 weeks with or without butyrate. $n = 6$ (Cre-/Ctrl), $n = 7$ (Cre-/Butyrate), $n = 5$ (Cre + /Ctrl), $n = 6$ (Cre + /Butyrate). **a** Weekly bodyweight changes in IL-22RA1$^{(f/f)}$ and IL-22RA1$^{Vil\ KO}$ mice with or without butyrate supplementation. **b** Food intake per mouse. **c** IPGTT in 16-h-fasted mice at 8 weeks. **d** The AUC during IPGTT. **e** ITT in 6-h-fasted mice at 10 weeks. **f** The AUC during ITT. **g** The levels of IL-22 from intestinal homogenates. **h** The serum levels of active GLP-1. **i** H&E staining of the mouse pancreas. Original magnification: 20× (scale bar,

50 μm). **j** Quantification of pancreatic islet size. Statistical significance was analyzed using two-way ANOVA (**c**, **e**) and Ordinary one-way ANOVA (**d**, **f**–**h**, **j**) with fisher's LSD test for multiple comparisons. $P < 0.05$ (*), $P < 0.01$ (**), $P < 0.001$ (***). Exact $P$ values shown in the figure correspond to comparisons between groups (Cre-/Ctrl) and (Cre-/Butyrate) (**c**, **e**). ns not significant. Data are presented as mean ± SEM. Box plots show the median (center line), the 25th and 75th percentiles (box), and the minimum and maximum values (whiskers). Representative data are shown from two independent experiments.

increase or dietary intake between the two groups (Fig. 5a, b). Glucose intolerance was alleviated in IL-22RA1$^{(f/f)}$ mice treated with butyrate, whereas no significant difference was observed in IL-22RA1$^{Vil\ KO}$ mice (Fig. 5c, d). Additionally, butyrate supplementation improved insulin resistance in IL-22RA1$^{(f/f)}$ mice. However, this improvement was not observed in IL-22RA1$^{Vil\ KO}$ mice (Fig. 5e, f).

We further evaluated IL-22 levels in intestinal tissues in this study. IL-22 levels increased in IL-22RA1$^{(f/f)}$ mice following butyrate supplementation, whereas no change in IL-22 levels was observed in IL-22RA1$^{Vil\ KO}$ mice regardless of butyrate supplementation (Fig. 5g). Consistent with these in vivo findings, ex vivo stimulation of lamina propria immune cells with butyrate increased IL-22 levels in IL-22RA1$^{(f/f)}$ mice, whereas LP cells from IL-22RA1$^{Vil\ KO}$ mice showed only minimal induction (Supplementary Fig. 14). Furthermore, plasma GLP-1 levels increased in IL-22RA1$^{(f/f)}$ mice, whereas no increase was observed in IL-22RA1$^{Vil\ KO}$ mice (Fig. 5h). In addition, butyrate supplementation increased the pancreatic islet size only in IL-22RA1$^{(f/f)}$ mice (Fig. 5i, j). HFD-fed IL-22RA1$^{Vil\ KO}$ mice showed reduced insulin-positive area, and butyrate increased insulin staining only in IL-22RA1$^{(f/f)}$ controls, but not in IL-22RA1$^{Vil\ KO}$ mice (Supplementary Fig. 15a, b). These results suggest that butyrate supplementation

contributes to IL-22-induced GLP-1 increase, thereby alleviating metabolic syndrome.

## IL-22 restored glucose intolerance in a GLP-1-dependent manner in obese mice

We investigated whether direct supplementation with IL-22, following butyrate-induced indirect IL-22 supplementation, could improve glucose intolerance in HFD-fed mice. One hour after IL-22 injection, serum GLP-1 levels were significantly elevated in RD-fed mice compared to PBS-treated controls (Fig. 6a). This indicates that IL-22 can stimulate GLP-1 production.

We further investigated whether impaired glucose intolerance could be directly alleviated by IL-22 treatment. IL-22 administration significantly improved glucose intolerance in IL-22RA1$^{f/f}$ mice but not in IL-22RA1$^{Vil\ KO}$ mice (Fig. 6b, c). Similarly, IL-22 treatment ameliorated insulin resistance in IL-22RA1$^{f/f}$ mice but not in IL-22RA1$^{Vil\ KO}$ mice (Fig. 6d, e). These results highlight the regulatory role of intestinal IL-22 signaling in glucose metabolism.

We treated HFD-fed IL-22RA1$^{f/f}$ and IL-22RA1$^{Vil\ KO}$ mice with the GLP-1 receptor agonist exendin-4 (Ex-4) every other day to determine whether GLP-1 supplementation could counteract the metabolic

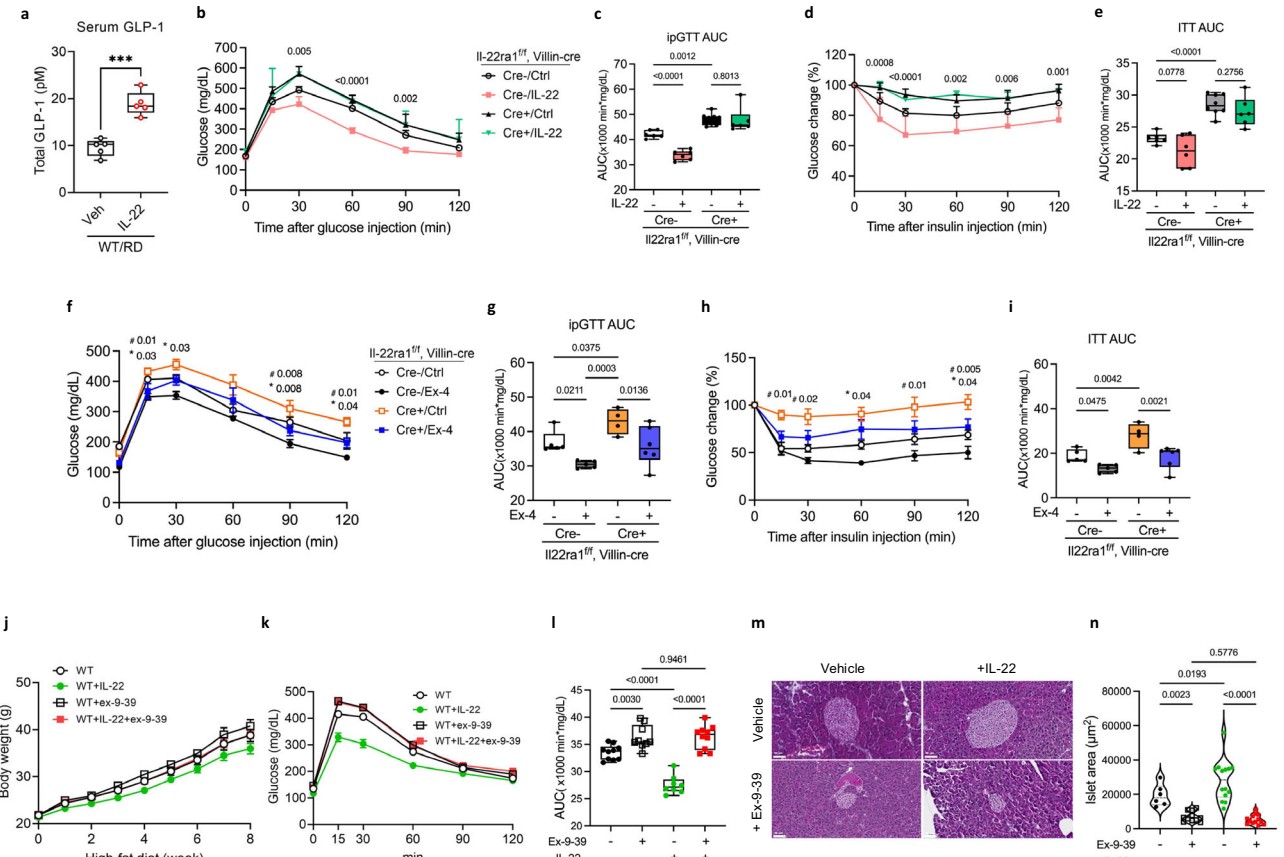

**Fig. 6 | IL-22 restores GLP-1 levels and ameliorates metabolic dysfunction. a** The serum level of GLP-1 secreted from mice fed a RD 1 h after injection of IL-22 (100 ng/mouse) (n = 4) (P = 0.0002). IL-22RA1$^{(f/f)}$ and IL-22RA1$^{Vil\ KO}$ mice were fed a HFD for 12 weeks with or without IL-22. For (**b**–**e**), n = 6 (Cre-/Ctrl), n = 6 (Cre-/IL-22), n = 8 (Cre +/Ctrl), n = 6 (Cre + / IL-22). **b** IPGTT in 16-h-fasted mice at 8 weeks. **c** The AUC during IPGTT. **d** ITT in 6-h-fasted mice at 10 weeks. **e** The AUC during ITT. IL-22RA1$^{(f/f)}$ and IL-22RA1$^{Vil\ KO}$ mice were fed an HFD for 12 weeks with or without Exendin-4. For (**f**–**i**), n = 5 (Cre-/Ctrl), n = 5 (Cre-/Ex-4), n = 4 (Cre +/Ctrl), n = 6 (Cre + / Ex-4). **f** IPGTT in 16-h-fasted mice at 8 weeks. **g** The AUC during IPGTT. **h** ITT in 6-h-fasted mice at 10 weeks. **i** The AUC during ITT. WT mice were subjected to an HFD for 8 weeks with recombinant IL-22 and ex-9-39. For (**j**–**n**), n = 10 (WT), n = 8 (WT + IL-22), n = 9 (WT + ex-9-39), n = 10 (WT + IL-22 + ex-9-39). **j** Weekly bodyweight changes with recombinant IL-22 or Ex-9-39 administration. **k** OGTT in 16-h-fasted mice at 8 weeks. **l** The AUC during OGTT. **m** H&E staining of the mouse pancreas. **n** Quantification of pancreatic islet size. Original magnification: 20× (scale bar, 50 μm). Statistical significance was analyzed using an unpaired two-tailed Student's t-test (**a**), two-way ANOVA (**b**, **d**, **f**, **h**, **j**, **k**), and Ordinary one-way ANOVA (**c**, **e**, **g**, **i**, **l**, **n**) with fisher's LSD test for multiple comparisons. P < 0.05 (*), P < 0.01 (**), P < 0.001 (***), P < 0.0001 (****). ns, not significant. Exact P values shown in the figure correspond to comparisons between groups (Cre-/Ctrl) and (Cre-/IL-22) (**b**, **d**). Asterisks (*) indicate statistical significance for comparisons between groups (Cre-/Ctrl) and (Cre-/Ex-4), and hash symbols (#) indicate statistical significance for comparisons between groups (Cre +/Ctrl) and (Cre + / Ex-4) (**f**, **h**). Data are presented as mean ± SEM. Box plots show the median (center line), the 25th and 75th percentiles (box), and the minimum and maximum values (whiskers).

syndrome exacerbated by IL-22 signaling deficiency. Ex-4 treatment significantly improved glucose tolerance in IL-22RA1$^{Vil\ KO}$ mice (Fig. 6f, g) and ameliorated insulin resistance compared to untreated controls (Fig. 6h, i). Ex-4 also increased the insulin-positive area in the pancreas in both IL-22RA1$^{(f/f)}$ and IL-22RA1$^{Vil\ KO}$ mice (Supplementary Fig. 16a, b). These findings indicate that GLP-1 reduction resulting from impaired IL-22 signaling contributes to metabolic dysfunction and that GLP-1 supplementation can mitigate these effects.

Finally, we co-administered IL-22 and the GLP-1 receptor antagonist exendin-9-39 (Ex-9-39) in HFD-fed WT mice to elucidate the interaction between IL-22 and GLP-1. IL-22 administration attenuated body-weight gain during HFD feeding, but this effect was abolished by co-administration of exendin-9-39 (Fig. 6j). Furthermore, the ability of IL-22 to regulate glucose intolerance in obesity was attenuated by Ex-9-39. This suggests that IL-22-mediated improvements in glucose tolerance are largely dependent on GLP-1 signaling (Fig. 6k, l). Furthermore, the increase in islet size induced by IL-22 was diminished in the presence of the GLP-1 receptor antagonist (Fig. 6m, n). These results underscore the pivotal role of IL-22 in the promotion of GLP-1 production and amelioration of metabolic dysfunction.

## Discussion

Our study identifies intestinal IL-22 signaling as an important regulator of GLP-1 production and glucose homeostasis in obesity. This finding expands upon previous evidence showing that the reduction of IL-22 expression is associated with the exacerbation of metabolic disorders. IL-22 supplementation regulates lipid metabolism in the liver and adipose tissue, thereby improving insulin sensitivity[20]. Despite extensive research on mucosal immune functions such as antimicrobial immunity and barrier integrity[22], the role of IL-22 in metabolic disorders remains unclear. Therefore, we investigated how IL-22 signaling impacts GLP-1 and metabolic regulation to clarify the association between IL-22 and metabolic disorders.

In our study, HFD-induced obesity is associated with reduced intestinal levels of GLP-1 and IL-22, which correlates with a decrease in IL-22-producing immune cells, specifically IL-22$^+$ ILCs and IL-22$^+$ CD4$^+$ T cells. This reduction in IL-22 levels, driven by an HFD, suggests a contributing role in the metabolic dysregulation observed in obese states. However, the specific mechanism underlying this decrease remains unclear.

The decrease in GLP-1 and IL-22 levels in HFD-fed mice is consistent with our analysis of the GSE datasets, which showed decreased expression of *GCG* and IL-22-related genes in patients with diabetes compared to non-diabetic patients with obesity. This positive correlation between GLP-1 and IL-22 expression suggests the disruption of a linked regulatory mechanism in metabolic disorders[25]. Moreover, the absence of IL-22 signaling in intestinal IL-22R-expressing cells led to significant systemic metabolic disturbances in our IL-22RA1[Vil KO] and IL-22RA1[Gcg KO] mouse models. These included worsened glucose intolerance and insulin resistance in HFD-fed mice. These metabolic defects were associated with reduced intestinal GLP-1 levels, decreased serum insulin concentrations, and impaired pancreatic islet function. Together, these results suggest that intestinal IL-22 signaling maintains glucose homeostasis and metabolic function, at least in part through its action on GLP-1.

GLP-1 production by intestinal L cells is regulated by various nutrient-sensing mechanisms. Carbohydrates, fats, and specific amino acids activate distinct pathways, including SGLT1-mediated glucose uptake, G protein-coupled receptor (GPCR) activation (e.g., GPR40 and GPR120), and T1R1/T1R3 heterodimers, respectively[4,26–29]. SCFAs, such as acetate, propionate, and butyrate, are produced through microbial fermentation of dietary fibers and act on GPCRs, such as GPR41 and GPR43 on L cells. This consequently enhances GLP-1 secretion through calcium signaling pathways[30,31]. Neural signals, including vagal nerve stimulation, contribute to GLP-1 release by releasing acetylcholine, which activates muscarinic acetylcholine receptors on L cells[32]. Inflammation and inflammatory factors, such as LPS and IL-6, regulate the release of GLP-1. Prolonged obesity and reduced IL-22 levels may further impair GLP-1 production, consequently disrupting its regulatory role in metabolic balance. Although previous studies have explored acute inflammation, the effects of chronic inflammation on metabolism warrant further research.

Our findings further demonstrate that IL-22 directly induces GLP-1 production in L cells, highlighting its critical role in this regulatory pathway. IL-22 may also influence other gut hormones promotes the production of PYY, another important metabolic hormone in the gut[33]. This suggests that IL-22 regulates the production of various metabolic hormones, including GLP-1 and PYY. The administration of IL-22 increased GLP-1 levels in both in vivo and in vitro models, demonstrating a dose-dependent effect. Furthermore, blocking calcium signaling or STAT3 activation attenuated IL-22–induced GLP-1 production, indicating that these pathways are important mediators of IL-22's effect on L cells.

In addition, our findings suggest that the gut microbiota regulates the IL-22–GLP-1 axis partly through SCFA-mediated induction of IL-22. The relationship between obesity and gut microbial dysbiosis is well established[34–36], and alterations in microbial composition can influence intestinal immune populations, including RORγt-expressing cells that are major sources of IL-22[18]. Consistent with this, HFD-fed mice in our study exhibited marked shifts in microbiota composition accompanied by decreased levels of SCFAs. SCFAs, particularly butyrate, are known to promote IL-22 expression in intestinal immune cells[23], and the reduced abundance of SCFA-producing bacteria likely contributes to the lower IL-22 and GLP-1 levels observed in obese mice. Notably, however, butyrate failed to increase IL-22 in IL-22RA1[Vil KO] mice. Ex vivo assays of LP immune cells showed that butyrate robustly induced IL-22 in LP cells isolated from IL-22RA1[(f/f)] mice but not in *Il22ra1*-deficient cells, indicating that epithelial IL-22RA1 signaling is required for effective SCFA-driven IL-22 induction. Together, these findings highlight the interplay between diet, the gut microbiota, epithelial IL-22 signaling, and intestinal immune regulation in maintaining metabolic homeostasis.

Although we did not detect a cell-intrinsic defect in IL-22 production in LP cells from IL-22RA1[Vil KO] mice, the blunted IL-22 response is likely secondary to altered mucosal homeostasis caused by impaired epithelial IL-22 signaling. Because IL-22 supports epithelial barrier integrity and local immune function, loss of epithelial IL-22RA1 may indirectly limit the ability of LP cells to respond to butyrate. Given that butyrate normally induces IL-22 in the gut[23], our findings also raise the possibility that butyrate-induced IL-22 participates in a positive feedback loop that reinforces IL-22 production under physiological conditions. Consistent with this, restoring IL-22 levels—either by butyrate supplementation or direct IL-22 administration—increased GLP-1 production and improved metabolic outcomes in obese IL-22RA1[(f/f)] mice, whereas IL-22RA1[Vil KO] mice showed no such improvement. These results indicate that the metabolic benefits of butyrate are largely mediated through IL-22–dependent pathways.

In line with this hierarchy, treatment with the GLP-1 receptor agonist Ex-4 restored glucose tolerance and insulin sensitivity even in the absence of intestinal IL-22 signaling, indicating that GLP-1 acts downstream of IL-22 to mediate its metabolic effects. Moreover, co-administration of IL-22 with the GLP-1 receptor antagonist Ex-9-39 attenuated the metabolic benefits of IL-22, further supporting that IL-22's glucoregulatory actions are largely mediated through GLP-1 signaling.

Additionally, our CCR6 knockout experiments revealed the importance of immune cell trafficking in the IL-22–GLP-1 axis. CCR6 is highly expressed on IL-22–producing immune cells such as Th17 cells, ILC3s, and lymphoid tissue inducer (LTi) cells[37–40]. The migration and localization of these RORγt+ cells to intestinal mucosal sites partially depends on CCR6[41]. Consistent with this, CCR6[-/-] mice had fewer IL-22–producing cells in the gut and significantly lower intestinal IL-22 levels than wild-type mice. CCR6 deficiency is also known to impair the recruitment of Th17 cells to Peyer's patches[42], which aligns with our observation of reduced IL-22 in CCR6[-/-] intestines, since Th17-derived IL-22 is crucial for gut health.

The decrease in IL-22 levels in CCR6[-/-] mice coincided with reduced GLP-1 levels and fewer GLP-1–expressing L cells in the intestine. GLP-1 is a critical incretin that enhances insulin secretion and maintains glucose homeostasis[4]. This reduction in GLP-1 likely contributed to the glucose intolerance observed in CCR6[-/-] mice, which exhibited impaired glucose tolerance than WT controls. Notably, long-term IL-22 administration in CCR6[-/-] mice restored GLP-1 production, improved glucose tolerance, and normalized insulin levels and pancreatic islet size. These therapeutic effects highlight the potential of IL-22 as a treatment for metabolic disorders characterized by GLP-1 deficiency and glucose intolerance.

Studies using IL-22R knockout (IL-22R KO) mice elucidated the multifaceted roles of IL-22. For instance, IL-22 signaling through IL-22R is essential for thermogenesis in white adipose tissue, as demonstrated in adipocyte-specific IL-22R KO mice. These mice exhibited reduced thermogenic gene expression and impaired thermogenesis under intermittent fasting conditions[43]. Kidney epithelial cell-specific IL-22R KO MRL/lpr mice exhibited significantly reduced renal inflammation and macrophage infiltration in lupus nephritis, indicating that IL-22 exacerbates disease pathology through its receptor-mediated activation of chemokine pathways[44]. Similarly, endothelial cell-specific IL-22R KO mice displayed decreased atherosclerotic plaque formation in cardiovascular research, highlighting the contribution of IL-22 to atherosclerosis and its potential as a therapeutic target[45]. Moreover, intestinal epithelial cell-specific IL-22R KO mice exhibited impaired gut epithelial repair, underscoring the importance of cytokines in the maintenance of intestinal integrity[43,46]. Additionally, IL-22RA1 deficiency in pancreatic beta cells induces age-dependent dysregulation of insulin biosynthesis and systemic glucose homeostasis[47]. This deficiency leads to increased cellular stress and inflammation in the islets, resulting in impaired insulin levels and ultimately causes glucose intolerance[47].

Collectively, our study demonstrates the critical role of IL-22 signaling in the enhancement of GLP-1 production and maintenance of

glucose homeostasis, though certain aspects remain unresolved. Systemic metabolic improvements observed with IL-22 supplementation suggest the involvement of specific immune pathways and signaling mechanisms, which were not fully delineated in this study. The observed reduction in GLP-1 levels likely reflects decreased hormone production, as supported by the reduced number of GLP-1–positive L cells and lower GLP-1 protein levels in both intestinal tissue and serum. An important question, however, is whether the impaired glucose tolerance in IL-22RA1–deficient mice arises primarily from diminished GLP-1 production or from defective β-cell adaptation to obesity. Under obese conditions, β-cells typically undergo compensatory hyperplasia to meet the increased insulin demand. Although whole-body Glp1r knockout mice maintain normal β-cell mass during metabolic stress[48], studies using GLP-1 agonizts and antagonists demonstrate that GLP-1 signaling can influence β-cell proliferation and survival[49–51]. Thus, reduced GLP-1 availability may contribute to the impaired β-cell adaptation observed in our IL-22RA1 KO models. However, the magnitude of β-cell loss in IL-22RA1-deficient mice under obese conditions exceeds what would be expected from GLP-1 deficiency alone, indicating the involvement of additional IL-22–dependent pathways. These findings underscore the need for further studies to delineate the relative contributions of impaired GLP-1 signaling and defective β-cell compensation to the altered islet phenotype in IL-22RA1-deficient mice.

Tissue-specific IL-22R KO models revealed local roles of IL-22 signaling in the intestine and pancreas but did not fully capture its broader systemic effects. Although Gcg-Cre also deletes IL-22RA1 in pancreatic α cells and IL-22 signaling has been reported in β cells, these islet-intrinsic pathways are unlikely to explain the metabolic phenotypes observed here. The reductions in GLP-1 production, impaired incretin responsiveness, and defective β-cell expansion are more consistent with loss of intestinal IL-22 signaling. Nonetheless, the relative contributions of intestinal and pancreatic IL-22 pathways to systemic insulin regulation remain to be clarified through future studies.

In summary, our study provides compelling evidence that IL-22 is an important mediator of GLP-1 production and that impaired IL-22 signaling contributes to the metabolic dysfunction observed in obesity. IL-22 supports glucose homeostasis and insulin secretion by enhancing GLP-1 production. Thus, it presents potential therapeutic avenues for treating metabolic disorders associated with obesity. HFD-induced changes in the gut microbiota, particularly a decrease in the abundance of SCFA-producing bacteria, reduced IL-22 production. Decreased IL-22 levels consequently reduce GLP-1 levels, thereby exacerbating metabolic disorders. Our findings indicate the protective role of IL-22 against obesity-induced metabolic dysfunction, suggesting its potential as a therapeutic target for addressing complications associated with obesity. The present findings align with previous research indicating that IL-22 maintains intestinal barrier integrity and modulates immune responses. Our study further demonstrates the direct impact of IL-22 on enteroendocrine cell function and glucose metabolism. Overall, IL-22-mediated GLP-1 regulates obesity-induced metabolic defects and prevents type 2 diabetes.

## Methods
### Mice
All experiments were approved by the Institutional Animal Care and Use committee of Kangwon National University (IACUC admission number KW-220808-1, KW-210826-2) and conducted in accordance with relevant ethical regulations. C57BL/6 background mice were purchased from Koatech (Pyeongtaek, Korea). Villin^Cre/+ (LML 115-449) mice were obtained from Asan Medical Center. Gcg^Cre/+ (MMRRC_042277-JAX), IL-22RA1^(f/f) (IMSR JAX:031003), and CCR6^−/− (IMSR JAX:005793) mice were purchased from the Jackson Laboratory.

GSK3 a/b f/f; RosaYFP + /-; Cg1 Cre mice were obtained from the College of Biomedical Sciences, Kangwon National University. Gcg^Cre/+ and IL-22RA1^(f/f) mice were crossed to induce L cell-specific deletion of IL-22RA1. Furthermore, Gcg^Cre/+ IL-22RA1(f/f) and ROSA-YFP mice were crossed to generate L cell-specific YFP-expressing reporter mice. Eight-week-old IL-22RA1^(f/f), Villin^Cre/+ IL-22RA1^(f/f), Gcg^Cre/+ IL-22RA1^(f/f), and CCR6^−/− male mice were fed either an RD (5L79; Orient Bio, Inc, Seongnam, Korea) or an HFD (60% of calories from fat, D12492; Research Diets, New Brunswick, NJ, USA) for 12 weeks. Food intake and body weight were measured weekly.

The mice were bred in specific pathogen-free conditions at the Animal Laboratory Center of Kangwon National University. Mice were intraperitoneally injected with PBS, IL-22 (20 μg/kg body weight; Biolegend, San Diego, CA, USA), Ex-4 (10 nmol/kg; Enzo life Sciences, Farmingdale, NY, USA), and Ex-9-39 (10 nmol/kg; Tocris, Bristol, United Kingdom) every other day for 10 weeks to evaluate the effects of IL-22, GLP-1 agonist, and GLP-1 antagonist. Moreover, mice were treated with butyrate (200 mM; Sigma-Aldrich, St. Louis, MO, USA) in drinking water for 14 weeks to investigate the effect of butyrate on IL-22 activation. The animals were kept in an animal facility at 20–22 °C, with 40–60% relative humidity and a 12-h/12-h (light/dark) cycle for at least 7 days before the experiment.

### Cell culture
STC-1 intestinal neuroendocrine tumor cells (American Type Culture Collection, Manassas, VA, USA) were cultured at 37.5 °C in Dulbecco's modified Eagle's medium (DMEM) supplemented with 5.5 mM glucose, 1% antibiotics, and 10% fetal bovine serum (FBS). The cultured cells were serum-starved for 4 h and then treated with IL-22 (0, 1, or 10 ng/mL) for 24 h. The total protein collected was stored at −80 °C until further use. Culture supernatants were collected on ice, clarified by centrifugation, mixed with DPP-IV inhibitor, and stored at −80 °C until analysis. Total GLP-1 levels in the supernatants were quantified using a Multi Species GLP-1 Total ELISA Bulk Kit (Merck, Darmstadt, Germany) according to the manufacturer's instructions.

### Chromatin immunoprecipitation assay
Chromatin immunoprecipitation (ChIP) was performed after identification of the STAT3 binding site in the mouse glucagon (GCG) promoter. This analysis was performed using a ChIP assay kit (Millipore Corp.). Briefly, cells were cross-linked in 1% formaldehyde solution for 10 min at 37 °C. Next, they were washed with PBS before incubation in lysis buffer (10 mM Tris-Cl [pH 8.0], 140 mM NaCl, 1 mM EDTA, 1% Triton X-100, 0.1% SDS, 0.1% deoxycholate, 1 mM $Na_3VO_4$, 1 μg/ml leupeptin, 1 μg/ml aprotinin, 10 mM NaF, and 0.2 mM phenylmethylsulfonyl fluoride). Cell lysates were then sonicated and centrifuged. The precleared supernatant was incubated with anti-STAT3 (Cell Signaling Technology, MA, USA; #9145, 1:100) or anti-IgG antibodies overnight at 4 °C. Immunoprecipitated DNA was amplified through PCR with primers corresponding to the GCG promoter region containing the STAT3 binding site. The primer pair sequences were: 5'-AGCACAAAACCCCTCAGTAAGT-3' (forward) and 5'-ATTTGTGAGGA-GAGCACTGGT-3' (reverse).

### Analysis of public RNAseq data
Bulk RNAseq data (GSE165512 and GSE132831) were downloaded from the GEO database; count data files of individual samples were then constructed as tables. Raw counts were normalized using the DESeq2 package. Next, data were analyzed using R statistical software (v4.1). Single-cell transcriptomic analysis reveals differential nutrient absorption functions in human intestines. In the present study, single-cell RNA sequencing data from the GSE125970 dataset were analyzed using R (v4.1) to investigate IL-22 receptor expression and perform cell clustering. The data were utilized to analyze the expression of IL-22 receptors and perform clustering.

## Isolation of lamina propria cells

Mouse small intestines were separated. Next, the feces were removed and intestines cut into pieces. The cut pieces were washed vigorously with PBS (Corning, NY, USA), treated with 1 mM DTT (Biosesang, Yongin, Korea)/PBS to remove the mucus, and shaken vigorously at room temperature for 5 min. The pieces were then incubated in FACS buffer (PBS supplemented with 2% FBS (Gibco, Waltham, MA, USA), 1% penicillin-streptomycin (P/S) (Gibco), 10 mM HEPES (Welgene, Gyeongsan, Korea), and 10 mM EDTA (Invitrogen, Vienna, Austria)) for 15 min at 37 °C. The samples were then washed thrice with fresh PBS before incubation in RPMI1640 (Corning) supplemented with 200 U/ml Collagenase IV (Worthington Biochemical, Lakewood, NJ, USA) and 100 µg/ml DNase I (Sigma-Aldrich) for 30 min at 37 °C in a shaker. Digested tissue was passed through a 70 µm nylon cell strainer (SPL Life Science, Pocheon, Korea). After centrifugation, the cells were suspended in 40% Percoll (Sigma-Aldrich) and layered on top of 70%. Centrifugation at room temperature was performed at $800 \times g$ for 20 min without a break. The layers of the interface between 40 and 70% were collected and washed with FACS buffer. The separated cells were prepared for flow cytometric analysis.

## Flow cytometry

Isolated lamina propria cells were stained with the following fluorescence conjugated anti-mouse antibodies in FACS buffer: Lineage Cocktail-APC (BD Biosciences, Bergen County, NJ, USA, #558074, 1:200)), CD45-FITC (BioLegend, #103108, 1:200), CD4-PerCP/Cy5.5 (BioLegend, #116012, 1:200), CD90.2-BV510 (BioLegend, #140319, 1:200), IL-7R-PE-Cy7 (BD Biosciences, #560733, 1:200), RORγt-BV421 (BD Biosciences, #562894, 1:200), and Live/Dead (Invitrogen).

All cells were stained for 20 min in the dark. Next, they were fixed and permeabilized with Fixation/Permeabilization Diluent (eBioscience, San Diego, USA) for intracellular staining. The data were acquired on a FACS Verse cytometer and analyzed using FlowJo software (both from BD Biosciences).

## Glucose and insulin tolerance tests

Glucose tolerance tests were performed at 8 weeks after HFD treatment in each group. The mice administered glucose either by oral gavage (1 g/kg) or intraperitoneal injection (1 g/kg) after 16 h of fasting. Blood samples were collected from the tail vein at 15, 30, 60, and 120 min after glucose injection.

Insulin tolerance was evaluated at 10 weeks. The mice were intraperitoneally injected with insulin (1 U/kg) after 6 h of fasting. Blood samples were then collected from the tail vein at 15, 30, 60, and 120 min after insulin injection. These samples were used to measure blood glucose concentrations using Auto-Check (APRILIS, Korea Co., Seoul, Korea).

## Glucose-stimulated insulin and GLP-1 measurement

The mice were fasted for 16 h and then administered glucose either orally (1 g/kg) or intraperitoneally (1 g/kg). At the designated time points, blood samples were collected from the retro-orbital plexus. Blood was collected into EDTA-coated tubes containing a DPP-4 inhibitor and immediately centrifuged at $1800 \times g$ for 10 min. Insulin levels were measured using the Mouse Ultrasensitive Insulin ELISA kit (Crystal Chem, IL, USA), and plasma active GLP-1 levels were measured using the Active GLP-1 (7-36) amide Chemiluminescence ELISA kit (ALPCO, NH, USA), according to the manufacturers' instructions.

## Intestinal crypt isolation and organoid culture

Mouse small intestinal tissue was cut into segments and washed in ice-cold PBS thrice. Mucus was removed from the intestinal surface through scratching using curved forceps. The cut small intestinal pieces were incubated in Cell recovery solution (Corning) for 30 min at 4 °C. Villi were carefully scraped with forceps and centrifuged at $80 \times g$

for 5 min. The villi were washed three times with Advanced DMEM/F12 (Gibco), and the intestinal crypts were isolated through pipetting with a 200 µl tip. Crypt cells were seeded on a 24-well plate embedded in 50% Matrigel (Corning) containing Advanced DMEM/F12. After gel formation, the wells were filled with organoid culture medium (Advanced DMEM/F12 supplemented with P/S, Glutamax (Invitrogen)) and HEPES with 50% L-WRN Media (Advanced DMEM/F12 supplemented with 20% FBS and P/S). B27 (Invitrogen), N2 (Invitrogen), N-acetylcystine (Sigma), mEGF (Invitrogen), and Y-27632 (Abmole, Houston, TX, USA)) were subsequently added. The organoids were cultured at 37 °C, and the medium was replaced every 2 days.

## Histological staining

Mouse ileum and pancreatic tissues were fixed in 10% neutral formalin, processed, and embedded in low-melted paraffin for histological examination. Paraffin-embedded tissue blocks were sectioned to 4 µm thickness and deparaffinized through immersion in xylene. Next, the samples were rehydrated by gradually reducing the ethanol concentration from 100% to 90% and then to 70%. The sections were stained using the H&E staining kit (Abcam, Cambridge, United Kingdom) according to the manufacturer's instructions.

## Immunofluorescence staining

Antigen retrieval was conducted in 10 mM sodium citrate buffer (Donginbiotech Co, Seoul, Korea) supplemented with 0.5% tween 20 (Duchefa Biochemie, Haarlem, Netherlands) for immunofluorescence staining. Blocking was conducted in 1% BSA (MP Biomedicals, Seoul, Korea) for 2 h at room temperature. The sections were stained with the primary antibody overnight at 4 °C. The secondary antibody was added and left to react for 2 h at 37 °C in the dark. It was then mounted with mounting solution containing 4,6-diamidino-2-phenylindole (DAPI) (Vector, Burlingame, CA, USA). Immunofluorescence images were captured using the LSM880 confocal laser scanning microscope (Carl Zeiss, Oberkochen, Germany) at the Central Laboratory of Kangwon National University. The samples were stained with the following primary antibodies: Mouse anti-GLP-1 (Novus Biologicals, Colorado, USA, #NBP2-23558AF488, 1:200), Recombinant Anti-Insulin antibody (Abcam, #EPR17359, 1:1000). The following secondary antibodies were used: anti-rabbit IgG (H + L)-Alexa 647 (Cell Signaling Technology, #4414, 1:500).

## Enzyme-linked immunosorbent assay

A 0.1-g mouse small intestine fragment was homogenized with a bead tube in 500 µl PBS. The homogenized tissue was centrifuged at $18,000 \times g$ for 10 min at 4 °C to obtain the supernatant. Intestinal IL-22 and total GLP-1 levels were subsequently measured using the OptEIA Mouse IL-22 Set (R&D Systems, Minneapolis, MN, USA) and Multi Species GLP-1 Total ELISA Bulk Kit (Merck). The collected blood samples were centrifuged at $1800 \times g$ for 10 min at 4 °C. Serum insulin and plasma active GLP-1 levels were measured using Mouse Ultrasensitive Insulin ELISA (ALPCO), Active GLP-1 (7-36) amide Chemiluminescence ELISA (ALPCO), and Glucagon ELISA Kit (Crystal Chem) according to the manufacturer's instructions. Absorbance was measured at 450 nm using SpectraMax i3 (Molecular Devices, San Jose, CA, USA).

## Metabolic cage analysis

Metabolic parameters were measured using the PhenoMaster metabolic monitoring system (TSE Systems, BH, Germany). Mice were individually housed in metabolic cages at 23–4 °C under a 12 h light/12 h dark cycle, with ad libitum access to food and water. After a 24 h acclimation period, measurements were recorded continuously for 48 h. Oxygen consumption ($VO_2$), carbon dioxide production ($VCO_2$), respiratory exchange ratio ($RER = VCO_2/VO_2$), and energy expenditure (EE) were calculated automatically using PhenoMaster software. EE was normalized to body weight.

### 16S rDNA amplicon sequencing

Fecal samples were collected from mouse intestines, preserved in DNA/RNA Shield (ZYMO RESEARCH, USA), and stored at −80 °C until further processing. Genomic DNA was extracted using the QIAamp Fast DNA Stool Mini Kit (QIAGEN, Germany) according to the manufacturer's instructions. The V4 region of the bacterial 16S rDNA gene was amplified using barcoded primers: forward primer V4_8F (5′-AGAGTTTGATCCTGGCTCAG-3′) and reverse primer V4_534R (5′-ATTACCGCGGCTGCTGG-3′). Sequencing was performed on the iSeq platform (Illumina, USA) with the iSeq Reagent Kit v3, generating 301 bp single-end reads. Quality control and microbiome data analysis were conducted using the QIIME2 pipeline (version 2023.05), and amplicon sequence variants were identified based on the SILVA database (version 138). Alpha diversity indices, including the Shannon, Gini-Simpson, and ACE, and beta diversity metrics (unweighted and weighted UniFrac, Bray-Curtis), were analyzed using QIIME2. Microbial composition and principal coordinate analysis (PCoA) plots were visualized in R (version 4.3.2). Statistical analysis of β-diversity was conducted using permutational multivariate analysis of variance to determine $p$-values.

### SCFA analysis

Fecal SCFAs, including acetic acid (Junsei Chemicals, Tokyo, Japan), butyric acid (Sigma-Aldrich Co.), and propionic acid (TCI, Tokyo, Japan), were quantitatively analyzed using gas chromatography-mass spectrometry (GC-MS). Fecal samples were collected from mice fed either an RD ($n = 7$) or an HFD ($n = 11$) and immediately stored at −80 °C until further analysis. Representative data are shown from two independent experiments.

Approximately 30 mg of fecal material was homogenized in 300 μL of ice-cold HPLC-grade water (J.T. Baker, Avantor Performance Materials, Center Valley, PA, USA) for 5 min at 40 oscillations per second using a TissueLyser LT (Qiagen, Hilden, Germany) with beads for SCFA extraction. The homogenized samples were centrifuged at $13,000 \times g$ for 10 min at 4 °C, and the supernatants were collected. The supernatant was acidified to pH 2 and extracted with diethyl ether (Daejung, Seoul, South Korea). The mixture was vortexed for 5 min, followed by centrifugation at $10,000 \times g$ at 4 °C. Next, the supernatant was combined with a derivatizing agent (BSTFA + TMCS, 1% TMCS, 99% BSTFA; Sigma-Aldrich Co.) at a 1:10 (v/v) ratio. An internal standard solution containing menthol (TCI) in diethyl ether (1:9, v/v) was also added. The resulting solution was incubated at 70 °C for 30 min to complete derivatization.

The derivatized sample (1 μL) was injected into the GC-MS system for analysis using a 7890 A GC system coupled with a 5975 C mass selective detector (Agilent Technologies, Santa Clara, CA, USA). SCFAs were separated on an HP-5 ms capillary column (30 m × 250 μm × 0.25 μm film thickness, Agilent Technologies), with helium (99.999% purity; Daejung, Seoul, South Korea) as the carrier gas (flow rate = 1.0 mL/min). The initial oven temperature was set to 30 °C for 3 min, increased to 120 °C at 15 °C/min for 1 min, and then raised to 250 °C at 50 °C/min for 3 min, yielding a total run time of 15.6 min. The injector temperature was set to 190 °C, and the ion source and quadrupole temperatures were maintained at 230 °C and 150 °C, respectively. Ionization was performed in electron impact mode at 70 eV. Finally, the analytes were quantified in selected ion monitoring mode with the target ions (m/z) for acetic acid, propionic acid, and butyric acid; the menthol internal standards were 117, 131, 145, and 123, respectively. GC–MS data acquisition and analysis were performed using MassHunter software (Agilent Technologies). Data processing and validation were conducted according to previously described methods[52].

### Total RNA extraction and quantitative real-time PCR

Total RNA was extracted from small intestinal tissue and intestinal organoid using TRIzol (Ambion, Austin, TX, USA). Complimentary DNA (cDNA) was synthesized using a cDNA kit (Promega, Madison, WI, USA), and RT-PCR was performed using the SYBR Green qPCR mix (Toyobo, Osaka, Japan) on the CFX96 Touch Real-Time PCR Detection System (Bio-Rad, Hercules, CA, USA). RT-PCR was performed using the following primers:

Gcg: (Forward 5′–AAGAGGAACCGGAACAACATTG–3′, Reverse 5′-GCCCTCCAAGTAAGAACTCACA-3′); RORγt: (Forward 5′-TGAGGCCATTCAGTATGTGG-3′, Reverse 5′-CTTCCATTGCTCCTGCTTTC-3′); Occludin: (Forward 5′-TGGCAAGCGATCATACCCAGAG-3′, Reverse 5′-CTGCCTGAAGTCATCCACACTC-3′); and Zo-1: (Forward 5′-GCTTTAGCGAACAGAAGGAGC-3′, Reverse 5′-TTCATTTTTCCGAGACTTCACCA-3′).

### Statistical analysis

All statistical analyses were performed using GraphPad Prism software (version 10 GraphPad Software Inc., San Diego, CA, USA). Comparisons between two groups were assessed using Student's t-test. In contrast, multiple comparisons were assessed using two- or one-way analysis of variance and multiple comparison tests. $P < 0.05$ (*), $P < 0.01$ (**), $P < 0.001$ (***), $P < 0.0001$ (****) indicate statistical significance.

### Reporting summary

Further information on research design is available in the Nature Portfolio Reporting Summary linked to this article.

## Data availability

The RNA sequencing datasets analyzed in this study are publicly available in the Gene Expression Omnibus (GEO) under the accession codes GSE165512 (unpublished), GSE132831[53], and GSE125970[54]. The generated sequencing data have been deposited in the NCBI Sequence Read Archive (SRA) under accession number PRJNA1392864. All other data supporting the findings of this study are available within the paper and its Supplementary Information. Source data are provided with this paper.

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

## Acknowledgements

This work was supported by the National Research Foundation of Korea (NRF) funded by the Ministry of Science and ICT (MSIT) of Korea (Grant

No. NRF-2020R1A5A8019180, RS-2024-00345147). We would like to thank Editage (www.editage.co.kr) for English language editing. Flow cytometry(Instrument No. NFEC-2022-09-281363) was performed at the Core-Facility for Innovative Cancer Drug Discovery (CFICDD) at Kangwon National University.

## Author contributions
C.-W.K. contributed to investigation, data curation, formal analysis, visualization, and wrote the original draft. J.-H.A. contributed to investigation, methodology, formal analysis, writing of the original draft, and review and editing. B.R.L. and H.M.K. conceptualized the study. Y.H. was responsible for investigation, validation, and resources. S.-E.K., H.J., J.-H.J., D.-J.K., Y.-B.L., S.M.K., H.H.Y., E.H.L., S.R.S., K.B.H., and E.S.L. conducted investigation. J.C. and J-K.K. contributed to data curation. H.P.K. and M.-N.K. contributed to writing—review and editing. S.-Y.C. contributed to data curation, formal analysis, visualization, and writing—review and editing. C.H.C. was responsible for conceptualization and writing—review and editing. H.-J.K. supervised the study, acquired funding, conceptualized the study, and contributed to writing—review and editing.

## Competing interests
The authors declare no competing interests.

## Additional information

Chae-Won Kim[1,11], Jae-Hee Ahn[1,11], Bo Ra Lee[1,2,11], Hong Min Kim[3,11], Youngjoo Han[1], Jae-Hyeon Jeong[1], Jaewon Cho[1], Hyunjin Jeong[1], Dae-Joon Kim[1], Seong-Eun Kim[1], Jeon-Kyung Kim[4], Yu-Bin Lee[4], Su Min Kim[5], Hye Hyun Yoo[5], Eun Hye Lee[6], Su Ryeon Seo[6], Kyung Bong Ha[3], Eun Soo Lee[3], Mi-Na Kweon[7], Hong Pyo Kim[8], Sun-Young Chang[8], Choon Hee Chung[3] ✉ & Hyun-Jeong Ko[1,9,10] ✉

[1]Department of Pharmacy, Kangwon National University, Chuncheon, Republic of Korea. [2]Preclincial Research Center (PRC), Daegu-Gyeongbuk Medical Innovation Foundation (K-MEDI hub), Daegu, Republic of Korea. [3]Department of Internal Medicine and Global Medical Science, Graduate School Wonju College of Medicine, Yonsei University, Wonju, Korea. [4]School of Pharmacy and Institute of New Drug Development, Jeonbuk National University, Jeonju, Republic of Korea. [5]Pharmacomicrobiomics Research Center, College of Pharmacy, Hanyang University, Ansan, Gyeonggi-do, Republic of Korea. [6]Department of Molecular Biosciences, Kangwon National University, Chuncheon, Republic of Korea. [7]Mucosal Immunology Laboratory, Department of Convergence Medicine, University of Ulsan College of Medicine/Asan Medical Center, Seoul, Korea. [8]Department of Pharmacy, and Research Institute of Pharmaceutical Science and Technology (RIPST), Ajou University, Suwon, Gyeonggi-do, Republic of Korea. [9]Innovative Drug Development Research Team for Intractable Diseases (BK21 Plus), Kangwon National University, Chuncheon, Republic of Korea. [10]Global/Gangwon Innovative Biologics-Regional Leading Research Center (GIB-RLRC), Kangwon National University, Chuncheon, Republic of Korea. [11]These authors contributed equally: Chae-Won Kim, Jae-Hee Ahn, Bo Ra Lee, Hong Min Kim. ✉e-mail: cchung@yonsei.ac.kr; hjko@kangwon.ac.kr

