## [Transparent Peer Review file · Nature Communications]

Intestinal interleukin-22 enhances GLP-1 production via the STAT3 pathway to improve glucose homeostasis in obese mice

Corresponding Author: Professor Hyun-Jeong Ko

Version 1:

Reviewer comments:

Reviewer #1

(Remarks to the Author)

Kim et al reported the involvement of IL-22 receptor signalling in mediating glucose homeostasis. Using *Vil-Cre* and *Gcg-cre* *Il22ra1* KO mouse models, they show that loss of *Il22R* signalling exacerbated glucose intolerance associated with lower GLP-1 levels and a reduced number of GLP-1+ cells in diet-induced obesity, with IL-22 treatment partially rescuing these phenotypes. In an enteroendocrine tumor cell line, IL-22 triggers pSTAT3 binding to the *Gcg* promoter. Reduced SCFA, particularly butyrate, was associated with the reduced IL-22 and GLP-1 observed in HFD-fed mice, and butyrate supplement enhances glucose tolerance and enhances islet size in HFD-fed WT but not *Vil1-Cre Il22ra1* KO mice. Exendin-4 but not IL-22 improved the glucose and insulin tolerance in *Vil1-Cre Il22ra1* KO mice.

The authors did an excellent job in interrogating multiple mouse models and demonstrating consistent metabolic phenotypes across. I am convinced that intestinal IL-22R signalling contributes to the maintenance of glucose homeostasis during diet-induced obesity. The data also agrees with previous data on IL-22 in maintaining metabolic health (PMID 39317186). However, the association between these changes and GLP-1 is weak, as the decrease in L cell numbers is unlikely to explain the apparent decrease in beta cell mass as reported in GLP-1R KO mice previously (PMID 22182839). This also agrees with the lack of effects in the exendin-9-39 blockade experiments on IL-22, which indicates that the GLP-1R signalling is not necessary for IL-22 to improve glucose tolerance.

Major comments

1. The data presented does not clarify the mechanism that causes glucose intolerance in the *Il22ra1* KO mice, whether it is by decreased GLP-1+ cell numbers, decreased GLP-1 production (does not imply secretion defects), or defective beta cell compensation. Notably, whole body *Glp1r* KO mice do not exhibit defective beta cell compensation post-HFD feeding (PMID 22182839), meaning that the change in L cell numbers is unlikely to explain the lack of beta cell compensation. This is also reinforced by the data in Fig 7j to 7i where IL-22 still works in the presence of exendin-9-39. In other words, the GLP-1 phenotypes are unlikely to relate to the defective beta cell compensation.
 - a. The authors should assess oral glucose tolerance by OGTT. If the decrease in GLP-1+ cells/GLP-1 content is relevant to the impaired ip glucose tolerance, then OGTT would reveal a similar difference. If there is no difference in oral glucose tolerance, the observed changes are likely driven primarily by reduced islet size. These experiments should be conducted in the *Vil-Cre* and/or *Gcg-Cre Il22ra1* KO mouse models.
 - b. There is a lack of functional assessment of GLP-1 secretion in vivo. The authors should measure glucose-dependent changes in serum GLP-1 levels during oral GTTs in metabolic studies whenever possible.
 - c. The authors should assess GLP-1 secretion in vitro in Figure 4. Instead of quantifying GLP-1 signals by imaging, they should measure GLP-1 secreted into the supernatant by STC-1 cells.
 - d. Please comment whether the L cells in the *Il22ra1* KO models decrease in absolute numbers or only downregulate GLP-1 expression.
 - e. The islet size phenotype is intriguing, as it suggests that in the *Il22ra1* KO mice, beta cells fail to adapt to HFD feeding which contributes to the glucose intolerance. Please quantify the insulin+ area for estimating beta cell area for Figs 2j, k, l, 3l/m/n, 6i/j and 7i. Also, is the pancreas weight different in the *Vil-Cre* and *Gcg-Cre Il22ra1* KO mice vs control post-HFD feeding?

- f. The authors should discuss the limitations of attributing the findings to the failure of beta cell mass to compensate for HFD feeding, vs the decreased L cell numbers/GLP-1 production, as the mechanism that drives glucose intolerance.
2. There is a lack of a functional assessment of insulin secretion in vivo to bridge between the observed GLP-1/beta cell dysfunction and glucose intolerance. Also in Fig 3k, the insulin levels are extremely low even in control mice. Normal fasting levels of circulating insulin in mice is approximately 0.5 ng/mL, with a couple of folds higher in HFD-fed mice. The authors should measure insulin levels at time 0 and 10 minutes after glucose administration in the oral and ipGTT experiments.
3. I do not agree with the interpretation of the results on the exendin-9-39 experiments in Fig 7, as IL-22 clearly works even in the presence of the antagonist. Please plot the area under the curve (AUC) for Figure 7k, quantify the islet area and size in Figure 7l, and perform proper statistical analysis.
4. Fig 3. Do these mice exhibit lower fasting circulating glucagon levels?
5. Supp Fig 3, does chronic IL-22 treatment increase L cell numbers on chow or HFD-fed WT mice?
6. Figs 2 and 3. Do these mice show differences in energy expenditure and body composition?
7. Figs 2 and 3. Is Il22ra1 specifically knocked down in the intestine of these mice? The Gcg-Cre targets pancreatic alpha cells as well; did the authors assess Il22ra1 expression in the pancreatic islets of these mice?
8. Fig 4. The mechanism of IL-22-mediated STAT3 binding to the glucagon promoter suggests that alpha cells can also be activated in a similar way. Is this described mechanism specific to L cells, or is it also relevant to alpha cells?
9. Figs 1 and 5 primarily present correlative data, which, while informative, do not significantly advance the manuscript's key arguments. These types of analyses might be more appropriate for human data. Data such as those from the Ccr6^{-/-} mice is more valuable to be included in the main figure.
10. Please comment on why butyrate treatment increases ileal IL-22 content from WT but not Villin-Cre Il22ra1 KO mice, considering that the line targets the IL-22 receptor not the cytokine itself.
11. Fig 7b/c and Supp Fig 6d. The Il22ra1 KO mice without IL-22 treatment show glucose intolerance in 7c when compared to control, but in Supp. Fig 6d, KO mice are not glucose intolerant when compared to control. This discrepancy is also seen in the insulin tolerance test (ITT) data in Figure 7d/e and Supp. Fig 6f, as well as between the Gcg-Cre mouse data in Figure 7f-i and Supp. Fig 7. The data in Supp. Figs 6 and 7 are based on n=3, which is likely underpowered compared to the data in the main figures.

Minor comments

1. There are formatting issues with the in-text citations
2. Line 66, "proglucagon" should be "preproglucagon" when referring to the gene
3. Line 74, "nonalcoholic fatty liver disease" should be replaced by "metabolic dysfunction associated liver disease"
4. Figure 2a/3a/6a/7j, please report the body weight of the animals as absolute values alongside the percentage changes as separate panels.
5. Figs 4b and 4c. Please specify how GLP-1 levels were measured in these experiments.
6. Please label whether total or active GLP-1 was measured in each applicable panel. Some panels (e.g. 6h) denote active but the others are not.
7. Line 429, "key" is a strong word, as IL-22 is only relevant to the ileal GLP-1 content in diet-induced obesity
8. Line 431, there is no data supporting changes in insulin "secretion" until it has been assessed.

Reviewer #2

(Remarks to the Author)

The manuscript by Kim et al. has described a new mechanism by which IL-22 improves organismal metabolism in obesity. The authors have provided evidence that IL-22 stimulated GLP-1 production in mice and that decreased short-chain fatty acids in HFD-fed mice led to reduced IL-22 and thus GLP-1. The study was well performed and is of great interest to the field. However, the reviewer has a few concerns that need to be addressed before the study becomes suitable for publication.

Major concerns:

- 1) In Fig. 2a, the authors show that Il22ra1-flox x Villin-Cre mice led to significantly more weight gain after HFD feeding, while no such changes can be observed in Il22ra1-flox x Gcg-Cre mice (Fig. 3a). How could this be explained and, if possible, please include this in the discussion. Furthermore, why are the metabolism-improving effects of IL-22 largely blocked by the GLP-1 antagonist ex9-39 in Fig. 7j?
- 2) It has been well established that short-chain fatty acids could stimulate GLP-1 secretion from L cells. However, it appears that butyrate fails to increase GLP-1 in Il22ra1-flox x Villin-Cre mice in Fig. 6h. Could the authors test whether the effect of butyrate on L cells is dependent on IL-22 signaling with cultured cells?
- 3) In Fig. 4f-g, the authors show that L cells account for ~ 10% of total cells by flow cytometry. However, immunostaining of organoids in Fig. 4h shows far fewer L cells among the total population. Please explain.
- 4) In Fig. 5, the authors describe changes of the microbiota and SCFA in HFD-fed mice, which have been well reported in the literature. Thus, related reports should be cited and the data could potentially be moved to supplemental data.
- 5) In Fig. 6g, why does butyrate fail to increase IL-22 levels in Il22ra1-flox x Villin-Cre mice?
- 6) In Il22ra1-flox x Villin-Cre and Il22ra1-flox x Gcg-Cre mice, the authors used ipGTT to examine metabolic dysfunction induced by HFD feeding. However, it could be more proper to perform oral GTT to examine the IL-22-GLP-1 axis.

Minor concerns:

- 1) Please make sure the whole figures are presented for a thorough review.
- 2) How are "IL-22-related genes" defined in this study, as used in 1i?
- 3) There is a data discrepancy between Fig. 2a vs Fig. 6a on the difference between Il22ra1-flox x Villin-Cre and Il22ra1-flox control mice. Please explain.
- 4) In ref.43, no changes in weight gains were observed between Il22ra1-flox x Villin-Cre and control mice. The authors

should acknowledge this and explain such a discrepancy.

Version 2:

Reviewer comments:

Reviewer #1

(Remarks to the Author)

A minor comment

1. Please use a proper representation of the dose of exendin-4 used (20 nM/kg corresponds to concentration not quantity).

Other than that, the authors have satisfactorily addressed my comments.

Reviewer #2

(Remarks to the Author)

The authors have successfully addressed my concerns and the manuscript is suitable for publication now.

Point-by-point response letter

Manuscript ID: NCOMMS-21-20352A-Z

Title: Intestinal interleukin-22 enhances GLP-1 production via the STAT3 pathway to improve glucose homeostasis in obese mice

We sincerely thank the reviewers and the editor for their constructive comments and suggestions, which have greatly improved our manuscript. Below, we provide a point-by-point response. Reviewer comments are shown in **bold italics**, and our responses follow. All changes in the revised manuscript are highlighted in the tracked version.

Reviewer #1 – Major Comment 1:

The data presented does not clarify the mechanism that causes glucose intolerance in the Il22ra1 KO mice, whether it is by decreased GLP-1+ cell numbers, decreased GLP-1 production (does not imply secretion defects), or defective beta cell compensation. Notably, whole body Glp1r KO mice do not exhibit defective beta cell compensation post-HFD feeding (PMID 22182839), meaning that the change in L cell numbers is unlikely to explain the lack of beta cell compensation. This is also reinforced by the data in Fig 7j to 7i where IL-22 still works in the presence of exendin-9-39. In other words, the GLP-1 phenotypes are unlikely to relate to the defective beta cell compensation.

Response:

We thank the reviewer for this insightful comment and the related questions (a–f). To address the reviewer's concerns more comprehensively, we conducted additional experiments during the revision. OGTTs in both Vil-Cre and Gcg-Cre Il22ra1 KO mice confirmed impaired glucose tolerance, accompanied by an attenuated postprandial rise in active GLP-1. In vitro studies using STC-1 cells and intestinal organoids further demonstrated that IL-22 directly enhances GLP-1 secretion. Consistent with these findings, immunostaining and Gcg mRNA analyses revealed a reduction in L-cell abundance. In parallel, quantification of insulin⁺ area across multiple cohorts showed a significant reduction in β -cell expansion, despite comparable pancreas weights.

In response to the reviewer's comment regarding the exendin-9-39 experiment, we repeated the study under identical conditions. The replicated data showed that IL-22 failed to improve glucose tolerance when GLP-1R signaling was pharmacologically blocked, confirming that IL-22 requires intact GLP-1R signaling to exert its glucoregulatory effects. We have revised the Results text and figure legend to clearly reflect this interpretation. This clarification has been added to the revised Discussion section (Line 423-426).

Importantly, although whole-body Glp1r KO mice maintain β -cell mass under HFD feeding¹, previous studies using GLP-1 agonists and antagonists^{2, 3, 4} indicate that GLP-1 can modulate β -cell proliferation and survival. Thus, impaired GLP-1 production may still contribute to the islet phenotype in our model. However, the magnitude of β -cell loss observed in Il22ra1 KO mice exceeds what would be expected from GLP-1 deficiency

alone.

Collectively, our findings support a model in which IL-22 regulates glucose homeostasis through both increased GLP-1 production and additional GLP-1-independent pathways, potentially involving modulation of β -cell stress or inflammatory signals. We have incorporated this interpretation into the revised Discussion and clarified the respective contributions of GLP-1-dependent and -independent mechanisms in the *Il22ra1* KO phenotype in response to the reviewer's constructive feedback.

a. The authors should assess oral glucose tolerance by OGTT. If the decrease in GLP-1+ cells/GLP-1 content is relevant to the impaired ip glucose tolerance, then OGTT would reveal a similar difference. If there is no difference in oral glucose tolerance, the observed changes are likely driven primarily by reduced islet size. These experiments should be conducted in the *Vil-Cre* and/or *Gcg-Cre* *Il22ra1* KO mouse models.

Response:

We thank the reviewer for this helpful suggestion. To directly address this point, we performed OGTT in both *Vil-Cre* and *Gcg-Cre* *Il22ra1* KO mice under identical dietary conditions. Consistent with our i.p. GTT data, both KO strains displayed significantly elevated glucose excursions compared with their respective controls, confirming that glucose intolerance is also exacerbated during OGTT. These findings support the conclusion that reduced GLP-1 signaling contributes to the impaired glucose tolerance, rather than the phenotype being explained solely by differences in islet size. These new results have been included in the revised figure (Fig. 2b, Fig. 2c, Fig. 3b, Fig. 3c).

b. There is a lack of functional assessment of GLP-1 secretion in vivo. The authors should measure glucose-

dependent changes in serum GLP-1 levels during oral GTTs in metabolic studies whenever possible.

Response:

We thank the reviewer for this insightful suggestion. In response, we measured serum active GLP-1 levels during oral glucose tolerance tests (OGTTs). In control mice, oral glucose administration led to an increase in circulating active GLP-1, whereas this glucose-dependent GLP-1 secretion was significantly reduced in both Vil-Cre and Gcg-Cre Il22ra1 KO mice. These results demonstrate that functional, glucose-dependent GLP-1 secretion in vivo is impaired in the absence of intestinal IL-22RA1 signaling (revised Fig.2d, Fig. 3d).

c. The authors should assess GLP-1 secretion in vitro in Figure 4. Instead of quantifying GLP-1 signals by imaging, they should measure GLP-1 secreted into the supernatant by STC-1 cells.

Response:

We thank the reviewer for this helpful suggestion. As requested, we quantified GLP-1 secreted into the culture supernatant. In the original (pre-revision) submission (Fig. 4b), ELISA data were already included but presented as values normalized to control, which may have caused confusion. To clarify, we have repeated the experiments and now present the newly acquired data as absolute ELISA values (pM). Consistent with our initial findings, IL-22 significantly increased GLP-1 secretion from STC-1 cells (revised Fig. 4b).

Moreover, to further strengthen this point, we also measured GLP-1 secretion in intestinal organoid cultures. Similarly, IL-22 treatment significantly enhanced GLP-1 secretion into the supernatant (revised Fig. 4i). These results consistently demonstrate that IL-22 directly promotes GLP-1 secretion in both enteroendocrine cell lines and intestinal organoids.

d. Please comment whether the L cells in the *Il22ra1* KO models decrease in absolute numbers or only downregulate *GLP-1* expression.

Response:

We thank the reviewer for this important question. In our *Il22ra1* KO models, intestinal *Gcg* mRNA levels (revised Figs. 2k and 3m) were significantly reduced, indicating a decrease in the overall abundance of GLP-1–producing cells. Consistent with this, immunostaining revealed a reduction in GLP-1⁺ L cells in the small intestine of *Gcg-Cre Il22ra1* KO mice, as determined by counting GLP-1–expressing cells (revised Fig. 3j and 3k). Because L cells were identified based on GLP-1 immunoreactivity, we cannot completely exclude the possibility that reduced GLP-1 expression per cell contributed to the lower counts. However, the concordant decrease in both *Gcg* transcript levels and GLP-1⁺ cell numbers supports the interpretation that L-cell abundance is reduced, rather than the phenotype being solely due to downregulation of GLP-1 within existing cells.

e. The islet size phenotype is intriguing, as it suggests that in the *Il22ra1* KO mice, beta cells fail to adapt to HFD feeding which contributes to the glucose intolerance. Please quantify the insulin⁺ area for estimating beta cell area for Figs 2j, k, l, 3l/m/n, 6i/j and 7i. Also, is the pancreas weight different in the *Vil-Cre* and *Gcg-Cre Il22ra1* KO mice vs control post-HFD feeding?

Response:

As requested, we quantified the insulin⁺ area as an estimate of β-cell area across the indicated cohorts. The results for the original Figs. 2j, 2k, 2l and 3l, 3m, 3n have been incorporated into the revised manuscript and are now presented in the updated Figs. 2n and 3p.

For the datasets associated with the original Figs. 6i, 6j, and 7i, we quantified the insulin⁺ area and include these results below for the reviewer’s reference (Supplementary Fig. 15a, b and Supplementary Fig. 16a, b).

We also measured pancreas weight post-HFD feeding and found no significant differences between Vill1-Cre or Gcg-Cre Il22ra1 KO mice and their littermate controls.

f. The authors should discuss the limitations of attributing the findings to the failure of beta cell mass to

compensate for HFD feeding, vs the decreased L cell numbers/GLP-1 production, as the mechanism that drives glucose intolerance.

Response:

We appreciate the reviewer's insightful comment. We agree that it is difficult to fully disentangle the relative contributions of reduced β -cell mass expansion and decreased GLP-1 production to the impaired glucose tolerance observed in Il22ra1 KO mice. In our models, deletion of IL-22RA1 signaling in intestinal epithelial cells and L cells results in reduced GLP-1 levels, as evidenced by decreased GLP-1⁺ L-cell numbers, lower intestinal and circulating GLP-1 protein levels, and reduced *Gcg* mRNA expression. We interpret this reduction in GLP-1 production as one contributor to the glucose intolerance phenotype.

At the same time, the degree of β -cell area reduction in Il22ra1 KO mice suggests that impaired β -cell adaptation to HFD feeding also plays a role. Because whole-body *Glp1r* knockout mice maintain β -cell mass under metabolic stress¹, the β -cell phenotype in our model likely involves additional GLP-1-independent mechanisms. Nevertheless, previous studies using GLP-1 agonists and antagonists^{2, 3, 4} show that GLP-1 can influence β -cell proliferation and survival, indicating that reduced GLP-1 availability may still contribute to the islet phenotype, although it is unlikely to be the sole determinant.

We have incorporated this clarification into the revised Discussion (lines 464-479) to more accurately reflect the limitations of attributing the glucose intolerance phenotype solely to either impaired β -cell compensation or reduced GLP-1 production. As described in the revised text:

The observed reduction in GLP-1 levels likely reflects decreased hormone production, as supported by the reduced number of GLP-1-positive L cells and lower GLP-1 protein levels in both intestinal tissue and serum. A key question, however, is whether the impaired glucose tolerance in Il22ra1 KO mice arises primarily from diminished GLP-1 production or from defective β -cell adaptation to obesity. Under obese conditions, β -cells typically undergo compensatory hyperplasia to meet the increased insulin demand. Although whole-body *Glp1r* knockout mice maintain normal β -cell mass during metabolic stress, studies using GLP-1 agonists and antagonists demonstrate that GLP-1 signaling can influence β -cell proliferation and survival. Thus, reduced GLP-1 availability may contribute to the impaired β -cell adaptation observed in our Il22ra1 KO models. However, the magnitude of β -cell loss in Il22ra1 KO under obese conditions exceeds what would be expected from GLP-1 deficiency alone, indicating the involvement of additional IL-22-dependent pathways. These findings underscore the need for further studies to delineate the relative contributions of impaired GLP-1 signaling and defective β -cell compensation to the altered islet phenotype in Il22ra1 KO.

Major Comment 2:

There is a lack of a functional assessment of insulin secretion in vivo to bridge between the observed GLP-1/beta cell dysfunction and glucose intolerance. Also in Fig 3k, the insulin levels are extremely low even in control mice. Normal fasting levels of circulating insulin in mice is approximately 0.5 ng/mL, with a couple of folds higher in HFD-fed mice. The authors should measure insulin levels at time 0 and 10 minutes after glucose administration in the oral and ipGTT experiments.

Response:

We assessed glucose-stimulated insulin secretion *in vivo* by measuring serum insulin levels at 0 and 15 minutes during both the OGTT and the ipGTT. Vil-Cre Il22ra1 KO mice exhibited significantly lower insulin responses compared with controls, indicating that impaired incretin signaling contributes to defective *in vivo* insulin secretion during the OGTT (revised Fig. 2e) and the ipGTT (revised Fig. 2f). Consistent with this, Gcg-Cre Il22ra1 KO mice also showed reduced insulin secretion during the OGTT (revised Fig. 3e). However, no significant difference in insulin release was observed during the intraperitoneal glucose tolerance test in this model (revised Fig. 3f).

To address the reviewer's concern regarding the low insulin levels in the original Fig. 3k, we re-measured the previously stored serum samples using a new ELISA batch (revised Figs. 2l and 3n). Although the values increased, they remained slightly lower than expected, likely due to partial insulin degradation during long-term storage. In contrast, newly collected samples measured with the same kit yielded insulin levels within the normal physiological range (revised Figs. 2e, 2f, 3e, and 3f), supporting the accuracy of the updated measurements.

Major Comment 3:

I do not agree with the interpretation of the results on the exendin-9-39 experiments in Fig 7, as IL-22 clearly works even in the presence of the antagonist. Please plot the area under the curve (AUC) for Figure 7k, quantify the islet area and size in Figure 7l, and perform proper statistical analysis.

Response:

We thank the reviewer for this important comment. To clarify, we repeated the exendin-9-39 experiments under the same conditions and included AUC analysis (revised Fig. 6l). In these replicated experiments, the glucose-lowering effect of IL-22 was not observed in the presence of exendin-9-39. This indicates that the metabolic

improvements of IL-22 are mediated through GLP-1 secretion and subsequent GLP-1R signaling. In addition, we quantified islet area and size in the corresponding cohorts (revised Fig. 6n). This clarification has been added to the revised Discussion (lines 403-406).

Major Comment 4:

Fig 3. Do these mice exhibit lower fasting circulating glucagon levels?

Response:

We thank the reviewer for this helpful suggestion. In response, we measured fasting circulating glucagon levels. Fasting glucagon concentrations were comparable between KO mice and their respective control littermates in both the Vil-Cre and Gcg-Cre Il22ra1 mice. These results indicate that impaired glucose tolerance in the KO mice is not associated with reduced glucagon levels. (revised Fig. 2q and Fig. 3s)

Major Comment 5:

Supp Fig 3, does chronic IL-22 treatment increase L cell numbers on chow or HFD-fed WT mice?

Response:

- We thank the reviewer for this question. In HFD-fed WT mice, chronic IL-22 treatment significantly increased the number of GLP-1⁺ L cells compared with untreated controls, indicating that IL-22 enhances L-cell abundance under metabolic stress. We have now added these results to the revised manuscript (Supplementary Fig. 3e, f).

Major Comment 6:

Figs 2 and 3. Do these mice show differences in energy expenditure and body composition?

Response:

We thank the reviewer for this comment. We assessed energy expenditure (EE) in both Vil-Cre Il22ra1 and Gcg-Cre Il22ra1 KO mice using metabolic cage analyses and did not detect any significant differences compared with their respective littermate controls (Supplementary Fig 7, 10).

Major Comment 7:

Figs 2 and 3. Is Il22ra1 specifically knocked down in the intestine of these mice? The Gcg-Cre targets pancreatic alpha cells as well; did the authors assess Il22ra1 expression in the pancreatic islets of these mice?

Response:

We thank the reviewer for this question. We did not directly quantify Il22ra1 deletion in each target tissue of the Vil-Cre and Gcg-Cre models and instead inferred tissue-specific deletion from the HFD-induced phenotypes. To examine deletion in pancreatic α cells of Gcg-Cre Il22ra1 mice, we performed confocal imaging of IL-22RA1. Although the antibody staining conditions were not optimal for high-resolution detection, we were still able to observe a noticeable reduction in IL-22RA1 signal in GCG⁺ cells from Cre⁺ mice compared with controls.

Major Comment 8:

Fig 4. The mechanism of IL-22-mediated STAT3 binding to the glucagon promoter suggests that alpha cells can also be activated in a similar way. Is this described mechanism specific to L cells, or is it also relevant to alpha cells?

Response:

We thank the reviewer for raising this important question. To address whether the IL-22–STAT3 binding mechanism observed in L cells is also applicable to α cells, we repeated the ChIP–qPCR experiment in parallel using identical conditions in STC-1 (L-cell model) and AlphaTC1 (α -cell model). As shown in the new panel, STC-1 cells displayed a clear IL-22–dependent increase in STAT3 binding to the Gcg promoter, consistent with our proposed mechanism. In contrast, AlphaTC1 cells did not show a similar induction; rather, STAT3 occupancy was reduced upon IL-22 stimulation, indicating that α cells respond in a distinct manner.

These results suggest that the IL-22–STAT3 regulatory mechanism described in our study is more specific to L cells and may not operate in the same way in α cells. However, the biological basis for the opposing pattern observed in AlphaTC1 cells is not yet clear. Further study will be required to clarify how IL-22 signaling influences α -cell function.

Major Comment 9:

Figs 1 and 5 primarily present correlative data, which, while informative, do not significantly advance the

manuscript's key arguments. These types of analyses might be more appropriate for human data. Data such as those from the Ccr6^{-/-} mice is more valuable to be included in the main figure.

Response:

We thank the reviewer for this constructive suggestion. We agree that the analyses in original Figs. 1 and 5 are primarily correlative and are best presented as supportive information rather than central to the mechanistic conclusions. Accordingly, we have moved these datasets to Supplementary Fig. 4 and 13.

Regarding the Ccr6^{-/-} data, we appreciate the reviewer's point; however, we believe that these results serve to complement the mechanistic framework rather than provide a key functional demonstration. As such, we feel that presenting the Ccr6^{-/-} dataset in the Supplementary Figures remains appropriate and maintains the clarity and focus of the main narrative.

Major Comment 10:

Please comment on why butyrate treatment increases ileal IL-22 content from WT but not Vil1-Cre Il22ra1 KO mice, considering that the line targets the IL-22 receptor not the cytokine itself.

Response:

We appreciate the reviewer's insightful comment. Butyrate is known to enhance IL-22 production by ROR γ ⁺ ILC3 and Th17 cells via SCFA receptors and HIF-1 α -dependent pathways⁵, and this effect is influenced by the intestinal microbiota. However, in our study, butyrate treatment did not increase ileal IL-22 levels in Vil-Cre Il22ra1 KO mice.

To investigate this discrepancy, we assessed the responsiveness of lamina propria (LP) immune cells to butyrate ex vivo and present these data in the Supplementary Figure 14. Butyrate markedly increased IL-22 secretion in LP cells from Il22ra1 Ctrl mice, whereas LP cells from Vil-Cre Il22ra1 KO mice showed minimal induction. These findings suggest that epithelial IL-22RA1 signaling is required for LP immune cells to mount an effective IL-22 response to butyrate.

Although our data do not demonstrate a cell-intrinsic defect in LP cells from Vil-Cre Il22ra1 KO mice, the reduced IL-22 induction is likely secondary to altered mucosal homeostasis resulting from loss of epithelial IL-22 signaling. Given IL-22's roles in epithelial barrier integrity, antimicrobial defense, and epithelial mediator production, disruption of epithelial IL-22RA1 signaling may indirectly impair LP immune responsiveness to butyrate. We have incorporated this clarification into the Discussion (lines 409–415).

Major Comment 11:

Fig 7b/c and Supp Fig 6d. The Il22ra1 KO mice without IL-22 treatment show glucose intolerance in 7c when compared to control, but in Supp. Fig 6d, KO mice are not glucose intolerant when compared to control. This discrepancy is also seen in the insulin tolerance test (ITT) data in Figure 7d/e and Supp. Fig 6f, as well as between the Gcg-Cre mouse data in Figure 7f-i and Supp. Fig 7. The data in Supp. Figs 6 and 7 are based on n=3, which is likely underpowered compared to the data in the main figures.

Response:

Response: We thank the reviewers for this careful comparison. These differences are primarily due to dietary differences rather than discrepancies within the same experimental conditions.

– **Original (pre-revision) Figure 7b/c vs Supplemental Figure 6d:** Original Fig. 7b/c (revised Fig. 6b/c) was obtained under HFD feeding conditions, and Il22ra1 KO mice consistently showed impaired glucose tolerance. In contrast, original Supplementary Fig. 6d (revised Supplementary Fig. 8d) was obtained from the regular diet cohort, and KO mice did not differ from controls. This is consistent with our model that intestinal IL-22RA1 signaling is important in HFD-induced metabolic stress.

– **Original (pre-revision) Figure 7d/e vs Supplemental Figure 6f:** A similar explanation applies to the ITT results. KO mice showed reduced insulin sensitivity under HFD conditions, but not under RD conditions (revised Supplementary Fig. 8f).

– **Original (pre-revision) Figure 7f-i vs Supp. Figure 7:** It should be noted that original Figures 7f-i represent Vil-Cre Il22ra1 KO mice on a high-fat diet, whereas original Supp. Fig. 7 (revised Supplementary Fig. 11) represents Gcg-Cre Il22ra1 KO on a RD with n > 10. Therefore, these datasets represent different models and dietary conditions, and the absence of a phenotype in original Supplementary Fig. 7 is due to RD rather than lack of power.

– **Sample size:** As mentioned previously, the cohort in original Supplementary Fig. 6 had n = 3 and was underpowered. Acknowledging the small sample size, we have repeated this experiment to expand the number of mice per group (revised Supplementary Fig. 8). Original Supplementary Fig. 7 (revised Supplementary Fig. 11) experiments already included adequate group sizes.

Taken together, these data suggest that impaired glucose and insulin tolerance were reproducibly observed under HFD conditions, whereas chow-only groups showed no overt intolerance, regardless of Cre driver, consistent with our working model.

Minor comments

1. There are formatting issues with the in-text citations

Response:

We appreciate the reviewer's comment regarding the formatting issues with the in-text citations. The formatting issues in the in-text citations have been corrected.

2. Line 66, "proglucagon" should be "preproglucagon" when referring to the gene

Response:

We thank the reviewer for catching this error. We have corrected “proglucagon” to “preproglucagon” when referring to the gene.

3. Line 74, “nonalcoholic fatty liver disease” should be replaced by “metabolic dysfunction associated liver disease”

Response:

As suggested, we have revised the terminology to “metabolic dysfunction–associated liver disease”.

4. Figure 2a/3a/6a/7j, please report the body weight of the animals as absolute values alongside the percentage changes as separate panels.

Response:

We thank the reviewer for this helpful suggestion. To fully address the comment, we include both the absolute body weight values and the percentage changes in the response letter. However, due to space and formatting constraints in the main manuscript, we have revised the relevant figures (revised Figs. 2a, 3a, 5a, and 6j) to present the absolute values.

5. Figs 4b and 4c. Please specify how GLP-1 levels were measured in these experiments.

Response:

We apologize for the lack of detail. GLP-1 levels in Figs. 4b and 4c were measured using a commercially available ELISA kit. We have added this information to the Methods section (Line 534-538).

6. Please label whether total or active GLP-1 was measured in each applicable panel. Some panels (e.g. 6h) denote active but the others are not.

Response:

We appreciate the reviewer's suggestion. We have now labeled each applicable panel to specify whether total or active GLP-1 was measured. For example, revised Fig. 6h shows active GLP-1, while revised Figs. 2i and 3i show total GLP-1. This clarification has been included in both the figure labels.

7. Line 429, "key" is a strong word, as IL-22 is only relevant to the ileal GLP-1 content in diet-induced obesity

Response:

We agree with the reviewer and have replaced "key" with "important" to provide a more accurate description.

8. Line 431, there is no data supporting changes in insulin "secretion" until it has been assessed.

Response:

We thank the reviewer for pointing this out. We agree that insulin secretion needs to be directly assessed to support this statement. In the revised manuscript, we have now measured glucose-stimulated insulin secretion in vivo by quantifying serum insulin levels at baseline and 15 minutes after oral glucose administration. Il22ra1 KO mice showed a significantly reduced insulin response compared with controls, confirming impaired insulin secretion. These new data have been added to the manuscript and the text has been revised accordingly.

Reviewer #2 (Remarks to the Author):

Major concerns 1:

In Fig. 2a, the authors show that Il22ra1-flox x Villin-Cre mice led to significantly more weight gain after HFD feeding, while no such changes can be observed in Il22ra1-flox x Gcg-Cre mice (Fig. 3a). How could this be explained and, if possible, please include this in the discussion. Furthermore, why are the metabolism-improving effects of IL-22 largely blocked by the GLP-1 antagonist ex9-39 in Fig. 7j?

Response:

We appreciate the reviewer's thoughtful comment. To clarify the variability in body weight, we evaluated this phenotype across four independent animal experiments, including the original pre-submission study and additional experimental sets generated during the revision. Although the first HFD-fed Villin-Cre Il22ra1 KO experiment showed a clear increase in body weight compared with controls, this phenotype was not reproduced in the subsequent three independent experiments. Across these repeated studies, body weight trajectories of Villin-Cre and Gcg-Cre Il22ra1 KO mice were comparable to those of their littermate controls. Based on these results, we conclude that body weight gain is not a consistent or reproducible phenotype associated with intestinal Il22ra1 deletion, and the initial finding likely reflects experiment-specific variation. This clarification has been added to the revised manuscript.

In contrast, the impairments in GLP-1 production and glucose-stimulated insulin secretion were robust and reproducible across all four independent experiments and in both tissue-specific knockout models. These findings indicate that defective incretin-islet signaling, rather than changes in body weight, represents the consistent metabolic consequence of intestinal Il22ra1 deficiency.

Regarding the reviewer's question about exendin-9-39, we repeated the experiment under identical conditions and performed AUC analysis (revised Fig. 6l). In this replicated study, IL-22 failed to improve glucose tolerance when GLP-1R signaling was blocked, demonstrating that IL-22's metabolic benefits require intact GLP-1R signaling. We also quantified islet area and size in the corresponding experimental set (revised Fig. 6n). These clarifications have been incorporated into the revised Results and Discussion (lines 423–426).

Major concerns 2:

It has been well established that short-chain fatty acids could stimulate GLP-1 secretion from L cells. However, it appears that butyrate fails to increase GLP-1 in Il22ra1-flox x Villin-Cre mice in Fig. 6h. Could the authors test whether the effect of butyrate on L cells is dependent on IL-22 signaling with cultured cells?

Response:

We thank the reviewer for this helpful suggestion. We performed additional experiments using intestinal organoid cultures. At higher in vitro concentrations (≥ 1 mM), butyrate alone modestly increased GLP-1 secretion, but co-administration of IL-22 resulted in a significantly greater increase. These results suggest that butyrate induces IL-22, which in turn promotes GLP-1 production, and that co-administration of IL-22 further enhances GLP-1 secretion. In vivo, butyrate was administered via drinking water (200 mM), but the

physiological concentrations reaching intestinal L cells are likely much lower. This may explain why butyrate alone did not raise GLP-1 in Vil-Cre Il22ra1 KO mice, whereas intact IL-22R signaling is required for effective GLP-1 induction under physiological conditions.

Major concerns 3:

In Fig. 4f-g, the authors show that L cells account for ~ 10% of total cells by flow cytometry. However, immunostaining of organoids in Fig. 4h shows far fewer L cells among the total population. Please explain.

Response:

This difference likely reflects methodological differences between the two approaches. Flow cytometry provides a quantitative assessment of all dissociated cells and may detect GLP-1⁺ cells more sensitively, resulting in an apparently higher proportion. By contrast, immunofluorescence images of intact organoids were obtained as z-stacks from representative regions, which give a qualitative impression and may underestimate the true frequency of L cells. Thus, the two approaches are complementary, with flow cytometry providing quantitative sensitivity and immunostaining reflecting spatial distribution.

Major concerns 4:

In Fig. 5, the authors describe changes of the microbiota and SCFA in HFD-fed mice, which have been well reported in the literature. Thus, related reports should be cited and the data could potentially be moved to supplemental data.

Response:

We thank the reviewer for this helpful suggestion. We have now cited the relevant literature and moved the microbiota and SCFA data to the Supplementary Fig. 13, as recommended.

Major concerns 5:

In Fig. 6g, why does butyrate fail to increase IL-22 levels in Il22ra1-flox x Villin-Cre mice?

Response:

We appreciate the reviewer's insightful comment. Butyrate is known to enhance IL-22 production by ROR γ ⁺ ILC3 and Th17 cells via SCFA receptors and HIF-1 α -dependent pathways⁵, and this effect is influenced by the

intestinal microbiota. However, in our study, butyrate treatment did not increase ileal IL-22 levels in Vil-Cre Il22ra1 KO mice.

To investigate this discrepancy, we assessed the responsiveness of lamina propria (LP) immune cells to butyrate *ex vivo* and present these data in the Supplementary Fig 14. Butyrate markedly increased IL-22 secretion in LP cells from Il22ra1 Ctrl mice, whereas LP cells from Vil-Cre Il22ra1 KO mice showed minimal induction. These findings suggest that epithelial IL-22RA1 signaling is required for LP immune cells to mount an effective IL-22 response to butyrate.

Although our data do not demonstrate a cell-intrinsic defect in LP immune cells from Vil-Cre Il22ra1 KO mice, the reduced IL-22 induction is likely secondary to altered mucosal homeostasis resulting from loss of epithelial IL-22 signaling. Given IL-22's roles in epithelial barrier integrity, antimicrobial defense, and epithelial mediator production, disruption of epithelial IL-22RA1 signaling may indirectly impair LP immune responsiveness to butyrate. We have incorporated this clarification into the Discussion (lines 409-415).

Major concerns 6:

In Il22ra1-flox x Villin-Cre and Il22ra1-flox x Gcg-Cre mice, the authors used ipGTT to examine metabolic dysfunction induced by HFD feeding. However, it could be more proper to perform oral GTT to examine the IL-22-GLP-1 axis.

Response:

We thank the reviewer for this helpful suggestion. To directly address this point, we performed OGTT in both Vil-Cre and Gcg-Cre Il22ra1 KO mice under identical dietary conditions. Consistent with our *i.p.* GTT data, both KO strains displayed significantly elevated glucose excursions compared with their respective controls, confirming that glucose intolerance is also exacerbated during OGTT. These findings support the conclusion that reduced GLP-1 signaling contributes to the impaired glucose tolerance, rather than the phenotype being explained solely by differences in islet size. These new results have been included in the revised figures (Fig. 2b, Fig. 2c, Fig. 3b, and Fig. 3c).

Minor concerns:

1) Please make sure the whole figures are presented for a thorough review.

Response:

We thank the reviewer for pointing this out. We have carefully checked the submission files to ensure that all figures are now presented in full for thorough review.

2) How are “IL-22-related genes” defined in this study, as used in 1i?

Response:

We apologize for the lack of clarity. In this study, “IL-22-related genes” refers to the following set of genes: REG3B, REG3G, SOCS3, FUT2, MUC1, SAA3, S100A8, LRG1, S100A9, SAA1, SAA2, IL10RB, RORC, AHR, STAT3. We have included this annotation in the legend of revised Supplementary Fig. 4b.

3) There is a data discrepancy between Fig. 2a vs Fig. 6a on the difference between $Il22ra1-flox \times Villin-Cre$ and $Il22ra1-flox$ control mice. Please explain.

Response:

We appreciate the reviewer’s careful comparison of original Fig. 2a and Fig. 6a. These panels were generated from independent HFD cohorts processed under the same protocol. A modest body-weight difference was observed in an initial cohort (original Fig. 2a), but this effect was not reproduced in subsequent cohorts, including the cohort shown in original Fig. 6a. In the course of revision, we repeated the measurements and the genotype effect on body weight was not significant. We have therefore updated Fig. 2a to reflect the current

dataset.

4) In ref.43, no changes in weight gains were observed between Il22ra1-flox x Villin-Cre and control mice. The authors should acknowledge this and explain such a discrepancy.

Response:

We thank the reviewer for this observation. While one of the initial cohorts contributed to the pooled significance in original Fig. 2a, other independent cohorts (e.g., original Fig. 6a) and additional sets performed during the revision did not reproduce this difference. Taken together, both our results and ref. 43 suggest that body weight is not a reproducible phenotype in this model. Instead, impaired glucose tolerance is consistently observed across cohorts and represents the major metabolic consequence of intestinal IL-22RA1 deficiency.

Reference

1. Lamont BJ, Li Y, Kwan E, Brown TJ, Gaisano H, Drucker DJ. Pancreatic GLP-1 receptor activation is sufficient for incretin control of glucose metabolism in mice. *J Clin Invest* **122**, 388-402 (2012).
2. Sridhar A, *et al.* Chronic exposure to incretin metabolites GLP-1(9-36) and GIP(3-42) affect islet morphology and beta cell health in high fat fed mice. *Peptides* **178**, 171254 (2024).
3. Wang Q, Brubaker PL. Glucagon-like peptide-1 treatment delays the onset of diabetes in 8 week-old db/db mice. *Diabetologia* **45**, 1263-1273 (2002).
4. Bregenholt S, *et al.* The long-acting glucagon-like peptide-1 analogue, liraglutide, inhibits beta-cell apoptosis in vitro. *Biochem Biophys Res Commun* **330**, 577-584 (2005).
5. Yang W, *et al.* Intestinal microbiota-derived short-chain fatty acids regulation of immune cell IL-22 production and gut immunity. *Nat Commun* **11**, 4457 (2020).

Point-by-point response letter

Manuscript ID: NCOMMS-21-20352B

Title: Intestinal interleukin-22 enhances GLP-1 production via the STAT3 pathway to improve glucose homeostasis in obese mice

We thank the reviewer for the comment on the revised manuscript. Below, we provide our responses to the remaining points raised.

Reviewer #1 – A minor comment 1:

1. Please use a proper representation of the dose of exendin-4 used (20 nM/kg corresponds to concentration not quantity).

Response:

We thank the reviewer for pointing this out. We apologize for the incorrect unit notation. The dose was mistakenly described as nM/kg, which is not appropriate for in vivo administration. Based on the actual dosing scheme, the administered dose corresponds to approximately 10 nmol/kg ($\approx 42 \mu\text{g/kg}$). The manuscript has been corrected throughout to reflect the appropriate units. This correction does not affect the experimental results or conclusions.